# Unveiling unconventional magnetism at the surface of $Sr_2RuO_4$

R. Fittipaldi[1,2,13], R. Hartmann[3,13], M. T. Mercaldo[2], S. Komori[4,10], A. Bjørlig[5], W. Kyung[6], Y. Yasui[7,11], T. Miyoshi[7], L. A. B. Olde Olthof[4], C. M. Palomares Garcia[4], V. Granata[2], I. Keren[8,12], W. Higemoto[9], A. Suter[8], T. Prokscha[8], A. Romano[1,2], C. Noce[1,2], C. Kim[6], Y. Maeno[7], E. Scheer[3], B. Kalisky[5], J. W. A. Robinson[4], M. Cuoco[1,2✉], Z. Salman[8✉], A. Vecchione[1,2] & A. Di Bernardo[3✉]

Materials with strongly correlated electrons often exhibit interesting physical properties. An example of these materials is the layered oxide perovskite $Sr_2RuO_4$, which has been intensively investigated due to its unusual properties. Whilst the debate on the symmetry of the superconducting state in $Sr_2RuO_4$ is still ongoing, a deeper understanding of the $Sr_2RuO_4$ normal state appears crucial as this is the background in which electron pairing occurs. Here, by using low-energy muon spin spectroscopy we discover the existence of surface magnetism in $Sr_2RuO_4$ in its normal state. We detect static weak dipolar fields yet manifesting at an onset temperature higher than 50 K. We ascribe this unconventional magnetism to orbital loop currents forming at the reconstructed $Sr_2RuO_4$ surface. Our observations set a reference for the discovery of the same magnetic phase in other materials and unveil an electronic ordering mechanism that can influence electron pairing with broken time reversal symmetry.

[1] CNR-SPIN, c/o University of Salerno, I-84084 Fisciano, Salerno, Italy. [2] Dipartimento di Fisica "E.R. Caianiello", University of Salerno, I-84084 Fisciano, Salerno, Italy. [3] Department of Physics, University of Konstanz, 78457 Konstanz, Germany. [4] Department of Materials Science and Metallurgy, University of Cambridge, Cambridge CB3 0FS, UK. [5] Department of Physics, Bar Ilan University, Ramat Gan 5920002, Israel. [6] Department of Physics and Astronomy, Seoul National University, Seoul 08826, Korea. [7] Department of Physics, Kyoto University, Kyoto 606-8502, Japan. [8] Laboratory for Muon Spin Spectroscopy, Paul Scherrer Institute, CH-5232 Villigen, PSI, Switzerland. [9] Advanced Science Research Center, Japan Atomic Energy Agency, Tokai, Ibaraki 319-1195, Japan. [10] Present address: Department of Physics, Nagoya University, Nagoya 464-8602, Japan. [11] Present address: RIKEN, Centre for Emergent Matter Science, Saitama 351-0198, Japan. [12] Present address: The Racah Institute of Physics, The Hebrew University of Jerusalem, Jerusalem 91904, Israel. [13] These authors contributed equally: R. Fittipaldi, R. Hartmann. ✉email: mario.cuoco@spin.cnr.it; zaher.salman@psi.ch; angelo.dibernardo@uni-konstanz.de

Electronic ordering in condensed matter systems often occurs as a result of a phase transition, meaning that it involves symmetry breaking. A classic example of the spontaneous breaking of time-reversal symmetry is ferromagnetism, which originates from the long-range ordering of electrons' spins.

In systems of reduced dimensionality like two-dimensional (2D) materials, the increase in quantum fluctuations compared to three-dimensional (3D) systems can induce symmetry-breaking phase transitions and quantum orders that do not have a 3D equivalent[1]. The emergence of topological phase transitions in 2D solids and 2D superfluids in the absence of standard long-range ordering were first proposed by Kosterlitz and Thouless[2], for which they were awarded the Nobel prize in 2016.

3D layered single crystals are the closest 3D analogue to 2D materials since electronic correlations in these crystals mainly develop inside the plane of each layer and the electrons' propagation is reduced along the crystal axis perpendicular to the layers. In layered single crystal oxide perovskites, the dominance of in-plane correlations between electrons of the $d$ orbitals often results in the emergence of exotic phases[3], some of which have been discovered over the past 30 years like high-temperature superconductivity[4], metal-to-insulator transitions[5] and multiferroicity[6].

$Sr_2RuO_4$ ($SRO_{214}$) is a peculiar oxide perovskite on the verge of various electronic instabilities that can be further stabilized by the asymmetry that the $SRO_{214}$ surface exhibits compared to the bulk, as a result of a structural reorganization of the surface $RuO_6$ octahedra. Apart from intense studies[7–9] aiming at determining the nature of the superconducting symmetry in $SRO_{214}$, which remains under debate, evidence for spin fluctuations[10] or magnetism under uniaxial pressure[11] has also been reported for $SRO_{214}$ single crystals in the normal state. These investigations, however, do not provide any information about the $SRO_{214}$ surface selectively but rather focus on the $SRO_{214}$ bulk properties.

At the surface of $SRO_{214}$, it has been theoretically suggested that conventional ferromagnetic ordering can emerge possibly stabilized by the rotation of the surface $RuO_6$ octahedra[12], but definitive evidence for the existence of magnetism at the $SRO_{214}$ surface has never been demonstrated, not even with scanning superconducting quantum interference device (SQUID) magnetometry[13]. Angle-resolved photoemission spectroscopy (ARPES) measurements on $SRO_{214}$ also reveal the presence of surface states[14], but the correlation between these surface states and magnetism is not conclusive[15].

Here, by using the extremely high sensitivity of low-energy muon spin spectroscopy (LE-µSR) to magnetic fields and its nanometre depth resolution[16,17], we find unambiguous evidence for the existence of an unconventional magnetic phase near the surface of $SRO_{214}$ single crystals. The hallmark features of the magnetic phase that we unveil in $SRO_{214}$ are a relatively high-temperature onset ($T_{on}$) between 50 and 75 K associated with a small amplitude of the magnetic moment (< 0.01 $\mu_B$/Ru atom, with $\mu_B = 9.27 \times 10^{-24}\,\mathrm{J\,T^{-1}}$ being the Bohr magneton), a homogeneous distribution of the sources of magnetism within the $ab$-plane of the $SRO_{214}$ crystals, and a decay in intensity of the magnetic signal from the $SRO_{214}$ surface over a length scale of ~10–20 nm. The features of this magnetic phase suggest that it cannot be reconciled with conventional ferromagnetism. We show instead that spin-orbital entanglement of the electronic states at the Fermi level results in orbitally-frustrated loop currents within the surface $RuO_6$ octahedra, which can generate the unconventional magnetism that we detect.

## Results

We perform LE-µSR measurements on $SRO_{214}$ single crystals grown by the floating zone method. The $SRO_{214}$ crystals used in this experiment are highly pure and have a superconducting critical temperature of ~1.45 K and residual resistivity ratio larger than 200, as evidenced by the X-ray diffraction and electronic transport measurements in Supplementary Fig. 1.

For the LE-µSR measurements, the crystals are cleaved with a non-magnetic $ZrO_2$ razor blade to avoid contamination from magnetic impurities and arranged to form a mosaic of size comparable to that of the muon beam (~2 cm in diameter) to maximize the amplitude of the signal (Fig. 1a). We perform most of the LE-µSR measurements with an external magnetic field ($\mathbf{B}_{ext}$) applied out-of-plane (i.e., along the $c$-axis of $SRO_{214}$) defined as the axis $\mathbf{z}$ of our orthonormal reference-axes system (Fig. 1b). The LE-µSR data are collected in two different configurations, namely both with the initial muon spin polarization vector ($\mathbf{S}_{\mu+}$) oriented perpendicular to $\mathbf{B}_{ext}$, known as transverse-field (TF) configuration and with $\mathbf{S}_{\mu+}$ collinear to $\mathbf{B}_{ext}$, known as longitudinal-field (LF) configuration (Fig. 1c). We also carry out zero-field (ZF) measurements in the same setup adopted for LF but with $\mathbf{B}_{ext} = 0$, as shown in Fig. 1c.

To determine the presence of any magnetism in $SRO_{214}$ and study its temperature ($T$) and depth dependence, we first perform LE-µSR temperature scans ($T$-scans) in the TF configuration as a function of energy ($E$). In the TF setup, the muons implanted with energy $E$ precess about the perpendicular $\mathbf{B}_{ext}$ at an average frequency $\omega_s(E) = \gamma_\mu B_{loc}(E)$ with $\gamma_\mu = 851.616\,\mathrm{MHz\,T^{-1}}$ being the muon's gyromagnetic ratio and $B_{loc}$ the amplitude of the local field experienced by muons. Each $E$ corresponds to a different muon implantation depth profile simulated using the Monte Carlo algorithm TrimSP[18] as shown in Fig. 1d.

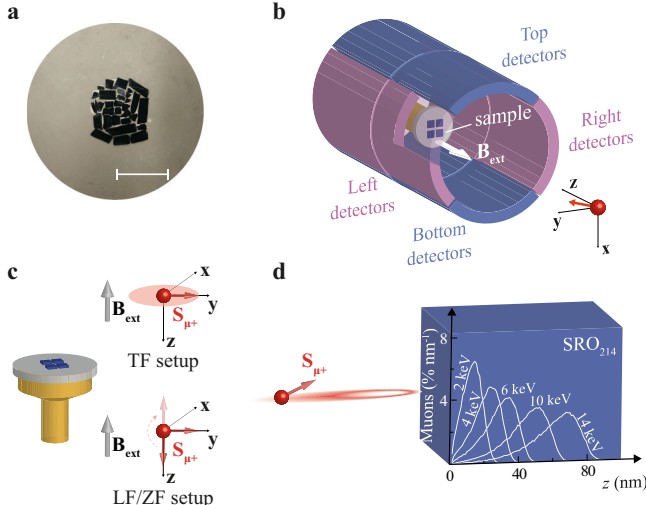

**Fig. 1 Low-energy µSR setup for measurements on $SRO_{214}$ single crystals. a** $SRO_{214}$ crystals cleaved and glued onto a Ni-coated aluminium plate to form a mosaic for the LE-µSR measurements. The scale bar corresponds to a length of 2 cm. **b** Experimental LE-µSR setup with applied field vector $\mathbf{B}_{ext}$ perpendicular to the sample (i.e., along the $c$-axis of $SRO_{214}$ coinciding with the axis $\mathbf{z}$ of our orthonormal reference-axes system) and arrays of positron detectors used to count muon decay events. The schematic cut-out allows viewing the sample inside the detectors. **c** LE-µSR measurement configurations for different orientations of the initial muon spin polarization vector $\mathbf{S}_{\mu+}$: $\mathbf{S}_{\mu+}$ perpendicular to the applied field vector $\mathbf{B}_{ext}$ and precessing in the $xy$-plane of our reference-axes system as indicated by the shadowed red circle (transverse field, top) or $\mathbf{S}_{\mu+}$ collinear to $\mathbf{B}_{ext}$ (longitudinal field or zero fields with $\mathbf{B}_{ext} = 0$, bottom). **d** Muon implantation profiles in $SRO_{214}$ simulated for a few representative implantation energies.

The $T$-scans are carried out whilst warming up the SRO$_{214}$ crystals, after zero-field cooling (ZFC) them and applying an external field with amplitude $B_{ext} = 100$ G at the lowest $T$. For the analysis of the $T$-scans in TF, we model the asymmetry signal $A_s(t)$ as $A_s(t) = A_0 e^{-\lambda t} \cos[\gamma_\mu B_{loc} t + \varphi_0]$, where $\lambda$ is the muon spin depolarization rate, which is proportional to the width of the local field distribution amplitude with average $B_{loc}$ sensed by muons, $A_0$ is the initial asymmetry which depends on the initial $S_{\mu+}$, and $\varphi_0$ is the initial phase depending on the initial $S_{\mu+}$ and on the geometry of the detector (see Supplementary Information). We note that the finite width of the muons' implantation profiles in LE-µSR (Fig. 1d) leads to a broadening of the field distribution experienced by the muons implanted at a given $E$. As a result, the asymmetry signal is better fitted assuming an exponential rather than a Gaussian relaxation rate, which is instead typically used in bulk-µSR studies where all muons implanted in a homogeneous sample experience the same field distribution. At a given $E$, we also perform a global fit[19] including all the data points collected as a function of $T$ and assuming $A_0$ and $\varphi_0$ as $T$-independent, since $A_0$ and $\varphi_0$ are both related to the initial $S_{\mu+}$ which is $T$-independent.

The results of the analysis are shown in Fig. 2, where we plot for each $E$ the $T$-dependence of the shift in $\lambda$, $\Delta\lambda(T)$, from the $\lambda$ value measured at the highest $T$ (~ 270 K in Fig. 2). The analysis of $\Delta\lambda$ allows to remove systematic effects such as variable contributions to $\lambda$ due to the measurement background.

An increase in $\Delta\lambda$ as $T$ is decreased signifies a broadening in the distribution of local fields experienced by muons at their implantation sites, and therefore it is a signature of enhanced magnetism emerging in the SRO$_{214}$ crystals as they are cooled down. Figure 2 shows that $\Delta\lambda$ increases as $T$ are lowered at all $E$s.

investigated, with a more pronounced increase in $\Delta\lambda$ occurring closer to the SRO$_{214}$ crystals' surface at $E = 3$ keV corresponding to an average muon stopping depth $\bar{z} \sim 15$ nm ($\bar{z}$ values are determined from the stopping profiles in Fig. 1d). The $\Delta\lambda$ values reported in Fig. 2, in combination with the corresponding raw asymmetry profiles and asymmetry fits reported in the Supplementary Information, show that $\Delta\lambda$ at $E = 3$ keV significantly changes slope at a $T$ between 50 and 75 K, which we identify as the $T_{on}$ of the magnetism. The data sets in Fig. 2 and Supplementary Fig. 7 for $E = 6$ keV and $E = 14$ keV also demonstrate that the onset temperature $T_{on}$ of the magnetism detected by muons decreases at higher implantation depths since $\Delta\lambda$ for $E \geq$ keV does not change significantly until a $T \sim 25$ K is reached, which is lower than the estimated 50 K < $T_{on}$ < 75 K. This result further confirms the surface nature of the magnetism that we measure in SRO$_{214}$ because the muons implanted deeper inside SRO$_{214}$ only experience an increase in their depolarization rate when the magnetism on the surface has become sufficiently strong, which occurs when $T$ has been decreased well below the onset of the magnetic phase transition at $T_{on}$.

We note that we have verified the reproducibility of the results reported in Fig. 2 and measured the same trends for $\Delta\lambda$ in two different batches of SRO$_{214}$ crystals, which demonstrates that the observed magnetism is an intrinsic property of SRO$_{214}$. The LE-µSR data on these two different batches of SRO$_{214}$ crystals have been collected over three beamtime sessions with various cryostats and magnets, which also rules out other possible artefacts related to the measurement setup. We confirm the emergence of magnetism from the $T$-dependence of $\Delta\lambda$ at different $E$ values also for a different TF configuration, where $B_{ext}$ is applied in-plane other than out-of-plane.

To further characterize the nature of the magnetic states observed in SRO$_{214}$, we study the response of these states in a higher applied $B_{ext}$ with amplitude $B_{ext} = 1500$ G in the TF setup (Fig. 3a, b). Although we do not observe significant variations in the local field amplitude, $B_{loc}$, with $T$ when $B_{ext} = 100$ G, Fig. 3a shows that with $B_{ext} = 1500$ G, $B_{loc}$ increases as $T$ is lowered for both $E = 3$ keV and $E = 14$ keV, before eventually decreasing in amplitude for $T < 25$ K. We also find that $B_{loc}$ at $E = 3$ keV deviates from that measured at $E = 14$ keV through exhibiting a positive shift from the latter for $T < 25$ K (Fig. 3a).

The $B_{loc}$ curves in Fig. 3a are reminiscent of the Knight shift determined by nuclear magnetic resonance (NMR)[10] in SRO$_{214}$—which is a measure of the local susceptibility or density of states of the material near the Fermi surface. The positive shift in $B_{loc}$ at $E = 3$ keV compared to $E = 14$ keV at $T < 25$ K (Fig. 3a) can therefore be correlated to an increase in the susceptibility based on the results in ref. [10], which is consistent with a strengthening of magnetism near the SRO$_{214}$ surface. The data in Fig. 3b also indicate that $\Delta\lambda$ in $B_{ext} = 1500$ G exhibits a clear increase at $E = 3$ keV within the same $T$ range reported in Fig. 2.

To determine the depth range of the magnetism whilst moving from the surface to the bulk of SRO$_{214}$, we perform energy scans ($E$-scans) for two different $B_{ext}$ amplitude values (100 and 1500 G). In Fig. 3c, d we report the depth variation of the shifts in the amplitude of $B_{loc}$, $\Delta B_{loc}$, and in $\Delta\lambda$ between $T = 5$ K and $T_{on}$ measured for $B_{ext} = 100$ and 1500 G. Although $\Delta B_{loc}$ versus $E$ is ~0 for $B_{ext} = 100$ G, in a higher applied $B_{ext} = 1500$ G we observe an increase in $\Delta B_{loc}$ for $E < 4$ keV up to ~0.65 G (corresponding to ~0.45% of $B_{ext}$). This result suggests that a $\Delta B_{loc}$ increase can also be present for $B_{ext} = 100$ G, but it may not be resolved, as it would fall within the experimental noise level of the LE-µSR technique. For both $B_{ext}$ values, $\Delta\lambda$ increases for $E < 4$ keV (Fig. 3d), meaning that, as the samples are cooled down, the magnetic signal probed by muons becomes stronger only up to an average implantation depth $\bar{z}$ of ~20 nm from the surface of the

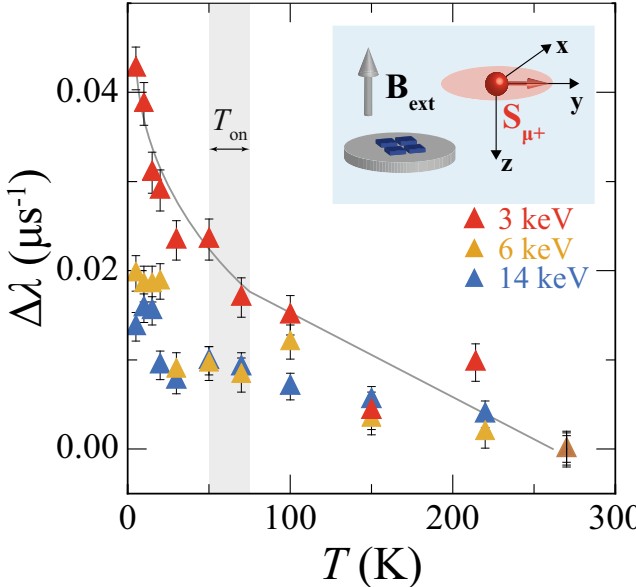

**Fig. 2 Temperature dependence of magnetism in SRO$_{214}$ at different implantation depths.** Shift in the muon depolarization rate, $\Delta\lambda$, from the $\lambda$ value measured at $T = 270$ K as a function of temperature $T$ measured in a TF setup (inset) with applied field amplitude $B_{ext} = 100$ G at different implantation energy $E$ values: $E = 3$ keV (red symbols with error bars), $E = 6$ keV (orange symbols with error bars) and $E = 16$ keV (blue symbols with error bars). The solid grey line serves as a guide to the eye and marks the $T$ range (grey shaded region) where $\Delta\lambda$ changes slope for $E = 3$ keV, which we identify as the onset temperature $T_{on}$ of the magnetism in SRO$_{214}$. The inset shows the relative orientation of the applied field $B_{ext}$ with respect to the muon spin polarization $S_{\mu+}$ in our orthonormal reference-axes system for the TF configuration.

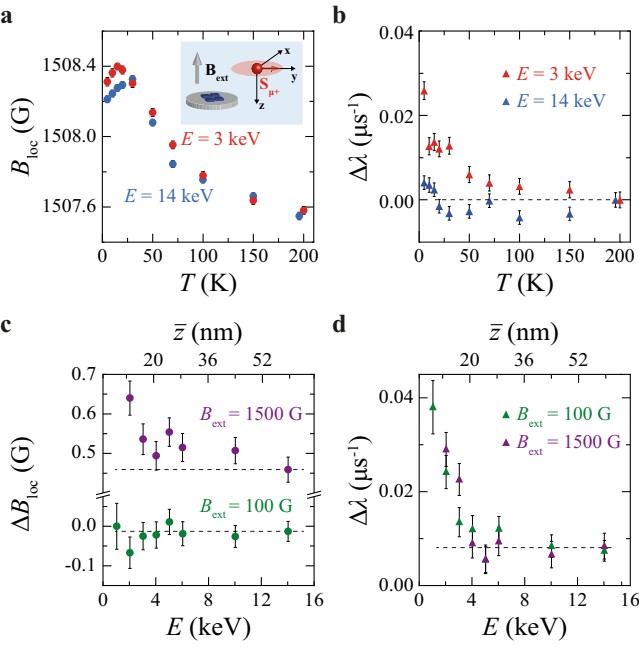

**Fig. 3 Magnetic field response and depth profile of magnetism in SRO₂₁₄.**
**a**, **b**, Temperature dependence of the local field amplitude $B_{loc}$ (**a**) and of the shift in the depolarization rate, $\Delta\lambda$, from the $\lambda$ value measured at $T = 200$ K (**b**) measured in TF with amplitude of the external field $B_{ext} = 1500$ G at $E = 3$ keV (red symbols with error bars) and $E = 14$ keV (blue symbols with error bars). Error bars in (**a**) are within the symbols. The inset schematic in (**a**) shows the relative orientation of the applied field $\mathbf{B_{ext}}$ with respect to the muon spin polarization $\mathbf{S_{\mu+}}$ in our orthonormal reference-axes system for the TF configuration. The $\Delta\lambda$ values in (**b**) are different from those shown in Fig. 2, as they are measured at a different stage of the experiment after warming the samples to room $T$, degaussing the magnet and zero-field cooling the samples again before applying $\mathbf{B_{ext}}$. **c**, **d** Shift in the local field amplitude $\Delta B_{loc}$ (**c**) and in $\Delta\lambda$ (**d**) between $T = 100$ K $> T_{on}$ and $T = 5$ K $< T_{on}$ measured in TF as a function of $E$ for $B_{ext} = 100$ G (green symbols with error bars) and $B_{ext} = 1500$ G (purple symbols with error bars). The top axes are the corresponding muon average stopping depth $\bar{z}$ values determined from the simulated muon stopping distributions in Fig. 1d. Dashed lines in (**b**) to (**d**) are guides to the eye.

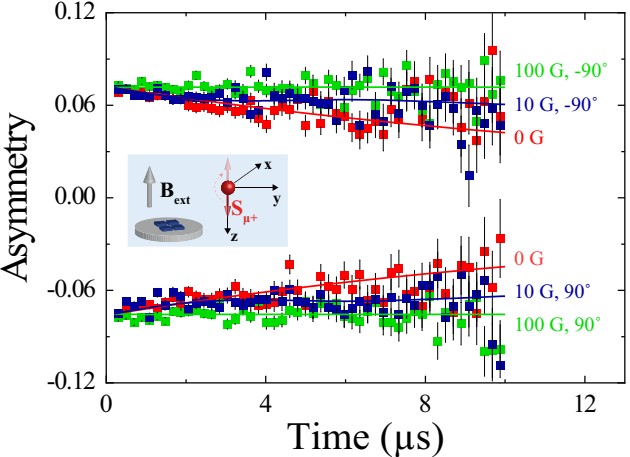

**Fig. 4 Static nature of magnetism in SRO₂₁₄.** Asymmetry signal measured at $T = 5$ K in ZF/LF for parallel (angle = 90°) and antiparallel (angle = −90°) alignments of the applied field $\mathbf{B_{ext}}$ and the muon spin polarization $\mathbf{S_{\mu+}}$ with different $B_{ext}$ amplitude values: $B_{ext} = 0$ G (red symbols with error bars), $B_{ext} = 10$ G (blue symbols with error bars) and $B_{ext} = 100$ G (green symbols with error bars). The inset schematic shows the relative orientation of $\mathbf{B_{ext}}$ and $\mathbf{S_{\mu+}}$ in our orthonormal reference-axes system for the ZF/LF setup.

SRO₂₁₄ crystals ($\bar{z}$ is determined from the simulated stopping profiles in Fig. 1d). This result also rules out magnetic impurities as the possible origin for the observed magnetic states because any magnetic impurities, if present, should not always be localized close to the surface of randomly cleaved SRO₂₁₄ crystals.

We note that a paramagnetic Knight shift in the ¹⁷O NMR signal has been recently measured for SRO₂₁₄ in its normal state under uniaxial strain[20]. The NMR Knight shift is of ~100 G in an applied field of $8 \times 10^4$ G, and it exhibits an anomalous enhancement related to spin fluctuations at the critical strain $\varepsilon_v$, defined as the strain value where the Fermi level reaches the Van Hove singularity (VHS). We note that in our experiment, the Fermi level of the SRO₂₁₄ surface layers is not at the VHS and the layers underneath are just bulk-like as demonstrated by previous ARPES measurements[21,22], whilst in ref. [20] the authors reach the VHS through the application of $\varepsilon_v$. Despite these dissimilarities between the two experiments, one can argue that the surface of SRO₂₁₄ has a different local strain compared to the bulk, meaning that there may exist a correlation between our $\Delta B_{loc}$ enhancement at the SRO₂₁₄ surface and the Knight shift. Drawing a quantitative comparison between our $\Delta B_{loc}$ enhancement and the Knight shift, however, is difficult for several reasons. First, as discussed in

ref. [20], the Knight shift includes several contributions which cannot be fully separated and it is specifically measured for the oxygen sites, whilst the muon stopping sites do not simply coincide with the oxygen sites, meaning that the interaction of the muons with SRO₂₁₄ is different. Second, our $\Delta B_{loc}$ shift (~0.2 G in a field of 1500 G) is smaller than the Knight shift reported in ref. [20] by several orders of magnitudes, and it is measured in different experimental conditions from those of the NMR experiment, which cannot be reproduced in the LE-μSR setup where neither larger magnetic field than those used, nor strain can be applied. These factors make it difficult to determine if and to which extent the shift in $\Delta B_{loc}$ would increase if the LE-μSR measurements could be done using similar settings to the NMR measurements. Last, even if we cannot exclude that a correlation between the NMR Knight shift and $\Delta B_{loc}$ exists, our LE-μSR measurements suggest that the $\Delta B_{loc}$ is characterized by different experimental signatures from those reported in ref. [20] for a paramagnetic Knight shift because $\Delta B_{loc}$ originates from an ordered phase that breaks time-reversal symmetry and that is also static in nature, as evidenced by our ZF measurements reported below.

We also perform measurements in an LF/ZF configuration to gain further insights into the nature of the magnetism observed in SRO₂₁₄ and in particular to determine whether the enhancement in $\Delta\lambda$ close to the SRO₂₁₄ surface for $T < T_{on}$ (Fig. 2) is due to an increase in static magnetic fields or to a reduction in spin fluctuations as $T$ decreases.

The measurements performed in the LF/ZF configurations demonstrate that the magnetism in SRO₂₁₄ is not related to spin fluctuations, but it rather has a static nature. We fit the LF/ZF asymmetry data to an exponential/Lorentzian Kubo–Toyabe function[23] (see also Supplementary Information).

Figure 4 shows that the damping in the asymmetry is decoupled as the applied field amplitude $B_{ext}$ is progressively increased from $B_{ext} = 0$ (ZF) to $B_{ext} = 100$ G (LF) for both directions of the collinear alignments of $\mathbf{S_{\mu+}}$ with $\mathbf{B_{ext}}$. The results clearly indicate that the local magnetic dipolar fields are static and of the order of ~10 G. If the asymmetry damping were instead due to magnetic fluctuations of the electronic moments, meaning due to spin-

lattice relaxation, a $B_{ext}$ of ~10 G would not affect the spin-lattice relaxation since the exchange energy due to Zeeman splitting in a field of ~10 G is much smaller than the thermal energy at $T = 5$ K.

To quantify the intensity of the static magnetism that we detect, we note that spin-polarized positive muons used in this experiment are likely to implant closer to an oxygen atom with the $SRO_{214}$ unit cell, due to the higher electron affinity of O compared to Ru. Since the Ru–O atomic bond length is ~2 Å in $SRO_{214}$ (ref. [24]), we estimate that the magnetic dipolar fields probed by muons at their implantation site (~10 G; Fig. 4) correspond to a magnetic moment much smaller than 0.01 $\mu_B$/ Ru atom.

## Discussion

To summarize our experimental findings, we detect a magnetic phase with 50 K < $T_{on}$ < 75 K (Figs. 2 and 3b) and localized within the first 10–20 nm from the $SRO_{214}$ surface (Fig. 3d), which induces a positive $\Delta B_{loc}$ shift near the surface only in higher $B_{ext}$ combined with a positive $\Delta\lambda$ shift independent on $B_{ext}$ (Fig. 3c, d). The magnetic phase detected is not related to fluctuations, but it is static in nature, and it corresponds to an average magnetic moment experienced by muons of <0.01 $\mu_B$/Ru atom at their implantation sites (Fig. 4). Based on these results, we can rule out several possibilities for the magnetic phases since they cannot account for our experimental observations.

The small magnitude of the moment in combination with the relatively high $T_{on}$ suggests that the magnetic phase detected at the $SRO_{214}$ surface is incompatible with conventional ferromagnetism. Ab-initio calculations indeed show that conventional ferromagnetic ordering of the Ru moments stabilized by the $RuO_6$ octahedra rotation at the $SRO_{214}$ surface would result in exchange energy due to Zeeman splitting of ~1 eV, which corresponds to a magnetic moment of ~1 $\mu_B$/Ru atom[12]. Conventional ferromagnetism due to surface $RuO_6$ octahedra distortion, which are present in our $SRO_{214}$ crystals after cleavage as confirmed by the low-energy electron diffraction measurements (LEED) in Supplementary Fig. 2, therefore cannot account for our LE-μSR results.

Similarly, single-unit-cell thick $SrRuO_3$ ($SRO_{113}$), which is the parent ferromagnetic compound of $SRO_{214}$, has a magnetic moment (~0.2 $\mu_B$/Ru atom, ref. [25]) that is a couple of orders of magnitudes larger than that probed by muons in our experiment and which can be detected by scanning SQUID magnetometry[25]. Supplementary Fig. 3 shows that scanning SQUID measurements performed by our group on the same crystals used for LE-μSR experiments cannot resolve any magnetic flux originating from $SRO_{214}$, which also rules out the presence of $SRO_{113}$ or other magnetic impurities in our $SRO_{214}$ samples, consistently with the depth dependence of magnetism in Fig. 3c, d. We note that, in our scanning SQUID measurements, we can only detect small magnetic spots on the $SRO_{214}$ crystals (Supplementary Fig. 3), most likely of extrinsic origin and possibly introduced during the cleaving process. These magnetic spots, however, only occupy a very small area of the sample surface (much smaller than 1%) and therefore they would only affect a small fraction of the implanted muons, meaning that they cannot account for the uniform increase in the depolarization rate measured in the LE-μSR signal below $T_{on}$. Our results obtained by scanning SQUID therefore further confirm that the magnetic signal which we resolve by LE-μSR in $SRO_{214}$ has to be intrinsic of the material and it is associated to a magnetic moment below the typical moment values expected for conventional ferromagnetism.

Further to conventional ferromagnetism, we also rule out magnetism due to spin textures with cancelling moments[26,27] or

to correlations between spurious magnetic impurities as a possible explanation for our results in $SRO_{214}$. This is because the appearance of such magnetic phases due to long-range correlations between magnetic spins or magnetic impurities embedded into a metallic Fermi sea at the relatively high onset temperature 50 K < $T_{on}$ < 75 K we measure would require a large strength of the Ruderman–Kittel–Kasuya–Yosida interaction and/or a strong crystal field anisotropy. This, however, should result in a magnetic moment much larger than the value that we measure (much smaller than 0.01 $\mu_B$/Ru atom).

We also exclude magnetic phases with antiferromagnetic ordering marked by a vanishing net magnetization and competing dipolar fields. This is because the $SRO_{214}$ is layered and tetragonal, and therefore has inequivalent distances between the in-plane neighbouring magnetic moments and the moments in adjacent $RuO_2$ planes. As a result, a muon implanted inside $SRO_{214}$ in any energetically favourable sites, for instance close to an apical oxygen due to its electrical affinity, would very unlikely experience an almost vanishing dipolar moment. Indeed, in ruthenates with antiferromagnetic properties, the $B_{loc}$ probed by muons corresponds to moments much larger than 0.01 $\mu_B$/Ru atom[28].

In addition to the above features, we also note that the time-reversal symmetry breaking (TRSB) normal (i.e., non-superconducting) phase that we detect at the $SRO_{214}$ surface has to be homogenously distributed within the $ab$-plane of the $SRO_{214}$ due to the monomodal local field distribution $p(B_{loc})$, as shown in the Supplementary Information. The magnetism sources should also correlate over a length scale comparable with the size of a single unit cell and be consistent with the $SRO_{214}$ translational symmetry. This is because the signal measured at a given $E$ is the sum of the contributions from all the muons implanted at $\bar{z}(E)$ in any position within the $ab$-plane of the $SRO_{214}$ crystals.

Possible unconventional normal-state TRSB phases which would meet the above requirements and generate weak static magnetic dipolar fields in the absence of long-range ferromagnetic ordering include intra-unit cell spin nematicity and electronic loop currents like those reported for other materials including iron-based superconductors[29,30], iridates[31] and cuprates[32,33]. The existence of a spin-nematic phase at the $SRO_{214}$ surface, however, can be excluded on the basis of symmetry considerations because a spin-nematic phase does not break inversion symmetry[34], and therefore it would be energetically unfavoured by the inversion asymmetric interactions occurring at the $SRO_{214}$ surface[35].

Our theoretical analysis reported below shows that the origin of the normal-state TRSB phase probed on the surface of $SRO_{214}$ can be ascribed to an orbital loop current with staggered magnetic flux. This orbital loop current phase is similar to that proposed to explain the intra-uni-cell antiferromagnetism in the pseudogap state of underdoped cuprates like $YBa_2Cu_3O_{6+\delta}$ and $HgBa_2$-$CuO_{4+\delta}$ (refs. [36,37]). The existence of an orbital loop current phase in cuprates, however, remains still controversial not only because earlier experimental evidence supporting the existence of this phase, and mostly based on spin-polarized neutrons[36,37], has not been confirmed by more recent studies, but also because alternative phases like charge density waves or spin density waves can equally account for the formation of the pseudogap in the normal state[38,39]. Similarly, for materials like iron-based superconductors, it is difficult to demonstrate conclusive evidence for an orbital loop current phase based on experiments demonstrating evidence for TRSB in the normal state because of the simultaneous presence of a TRSB spin density wave in the same materials[40]. To the best of our knowledge and as reported in ref. [41], the only two material systems for which evidence for an orbital loop current phase has been reported without effects that

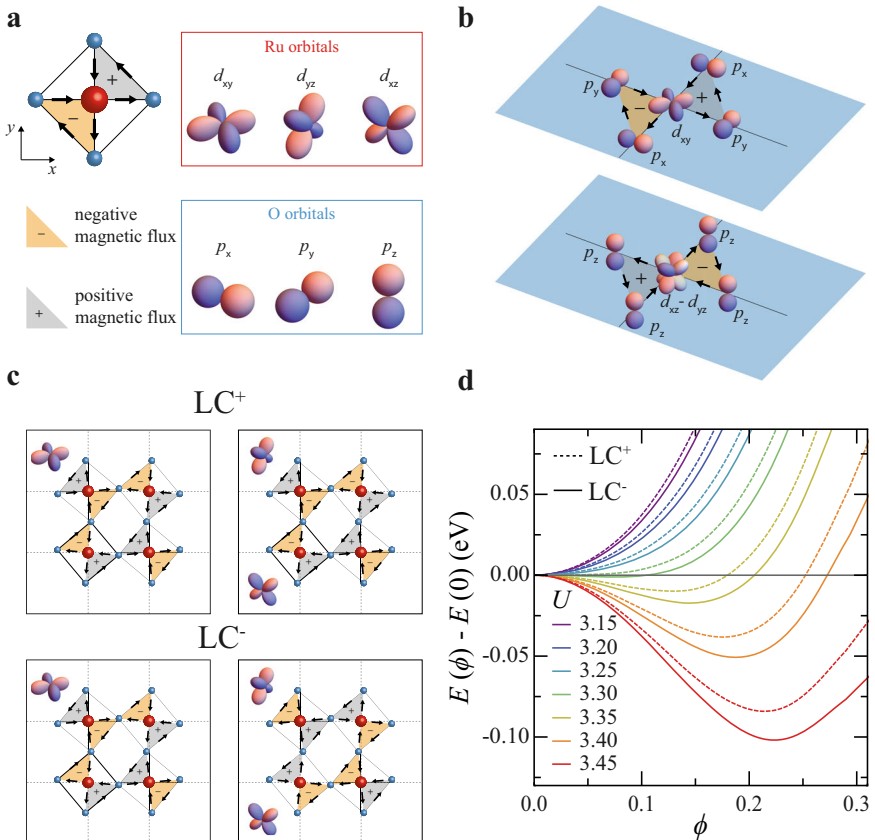

**Fig. 5 Magnetism due to orbital loop currents in SRO$_{214}$. a** Illustration of the RuO$_4$ plaquette and of the corresponding $d$-orbitals for the Ru atoms (red box) and $p$-orbitals for the O atoms (light blue box) with asymmetric loop current distributions generating magnetic flux pointing inward (yellow triangle with '−' symbol) or outward (grey triangle with '+' symbol) the RuO$_4$ plane. **b** Possible orbital loop currents for a given RuO$_4$ plaquette associated with the Ru–O hybridization of the $d_{xy}$ orbitals (top) and of the ($d_{xz}$, $d_{yz}$) orbitals (bottom). **c** Loop current states with equal (LC$^+$ state) and opposite sign (LC$^−$ state) of the magnetic flux associated with the $xy$- and $z$-orbital sectors of the RuO plaquettes in the SRO$_{214}$ supercell. **d**, Free energy $E(\phi) - E(0)$ of the LC$^+$ (dashed lines) and LC$^−$ states (solid lines) calculated at $T = 50$ K for different values of the $d–p$ Coulomb interaction $U$ ($U$ values given in eV units are specified in the figure legend) as a function of the order parameter $\phi$ which sets the amplitude of the bond current.

can be related to other coexisting TRSB phases include the Mott insulator Sr$_2$IrO$_4$ using non-linear optical microscopy[42] and the two-leg ladder cuprate Sr$_{14−x}$Ca$_x$Cu$_{24}$O$_{21}$ using spin-polarized neutrons[43]. Spin-polarized neutron studies, however, have not confirmed the presence of normal-state TRSB orbital loop currents in Sr$_2$IrO$_4$ (ref. [42]), whilst the orbital loop current phase reported for Sr$_{14−x}$Ca$_x$Cu$_{24}$O$_{21}$ cannot be directly correlated to that which we propose for SRO$_{214}$, since Sr$_{14−x}$Ca$_x$Cu$_{24}$O$_{21}$ has radically different physical properties in that it is non-superconducting and behaves like a spin liquid above a certain hole doping.

To understand the physical mechanism underlying the magnetism measured in SRO$_{214}$, we consider orbital loop currents emerging as a result of electronic instabilities at the Fermi level. We use a tight-binding description of the electronic structure of SRO$_{214}$ including $d$-orbitals at the Ru sites and $p$-orbitals at the planar O sites and consider $d–p$ and $p–p$ Coulomb interactions as responsible for the electronic instabilities yielding the orbital loop current phase (Fig. 5a).

Inversion symmetry breaking at the SRO$_{214}$ surface rules out an orbital loop current phase that is spatially symmetric like that consisting of orbital currents flowing along each bond of the RuO$_4$ plaquette (see Supplementary Information). Based on symmetry considerations, we therefore restrict our analysis to magnetic phases originating from asymmetric orbital loop

currents, namely combinations of clockwise and anticlockwise orbital currents generating opposite magnetic fluxes only within two sections of the RuO$_4$ plaquette (Fig. 5a) and determine whether this type of phase is consistent with the features of the magnetism observed in SRO$_{214}$.

For a given RuO$_4$ plaquette, there are two possible asymmetric TRSB loop current phases that differ by the way loop currents are distributed for the $d_{xy}$ and ($d_{xz}$, $d_{yz}$) orbitals sectors (Fig. 5b). Asymmetric loop current phases of several neighbouring plaquettes must combine at the SRO$_{214}$ surface compatibly with the rotation of the RuO$_6$ octahedra to yield the loop current distribution for a SRO$_{214}$ supercell. The net loop current distribution in the SRO$_{214}$ supercell has also resulted in staggered magnetic fluxes to prevent spontaneous current flow or charge accumulation, the existence of which is not possible in the metallic state of SRO$_{214}$.

The first result which we obtain from our calculations is that the magnetic phase originating from orbital loop currents flowing inside the SRO$_{214}$ supercell can indeed be the ground state of SRO$_{214}$ because this magnetic phase is characterized by lower free energy compared to the non-magnetic SRO$_{214}$ configuration. We also find that this unconventional orbital loop current (LC) phase can stabilize into two different energy states, the degeneracy of which is lifted by spin–orbit coupling in SRO$_{214}$, depending on the distribution of the loop currents around the $d$-orbitals at the

Ru sites. Figure 5c shows that one of these two magnetic states is generated by loop currents having the same sign for the $d_{xy}$ and $(d_{xz}, d_{yz})$ orbitals at the Ru sites, here denoted as LC$^+$ state, whilst the other state is generated by loop currents for the $d_{xy}$ and $(d_{xz}, d_{yz})$ orbitals at the Ru sites having opposite sign (LC$^-$ state).

A calculation of the free energy for the LC$^-$ and LC$^+$ states as a function of the Coulomb interaction $U$ between electrons of the $p$- and $d$-orbitals of the O and Ru atoms for $T$ below $T_{on}$ shows that the LC$^-$ state is energetically favourable compared to the LC$^+$ state (Fig. 5d and Supplementary Information). Figure 5d also shows that the free energy gain with respect to the normal state with zero flux is ~20–30 meV which is consistent with the order of magnitude of $T_{on}$. The order parameter $\phi$ in Fig. 5d is the expectation value of the asymmetric bonding operator in SRO$_{214}$ (see Supplementary Information).

We last verify that the orbital loop currents corresponding to the most stable LC$^-$ configuration are consistent with the strength of the magnetic field probed by muons near the SRO$_{214}$ surface. The magnetic field in the LC$^-$ configuration is obtained by determining the average current flowing along each bond inside the SRO$_{214}$ supercell and then deriving the net total field according to the Biot–Savart law. Using experimental values for the magnetic permeability of SRO$_{214}$, $\mu_m \cong 2 \times 10^{-2}$ G m$^{-1}$ A$^{-1}$ (ref. [10]), and ab-initio values for the nearest-neighbour hopping parameters $t$ for the $d$- and $p$-orbitals[35,44], we obtain a $B_{loc}$ amplitude in the range of 5–15 G which is consistent with the order of magnitude measured experimentally by LE-μSR.

In conclusion, our study demonstrates clear evidence for the existence of an unconventional magnetic phase in the surface region of SRO$_{214}$. We show that loop currents with vanishing net magnetic flux due to orbital frustration can account for the hallmark signatures of the observed TRSB phase at the SRO$_{214}$ surface. The orbital-dependent nature of the loop current phase suggests that mechanisms lowering the crystalline symmetry, e.g. strain, can increase the orbital imbalance and result in stronger magnetism. From this point of view, our results can be linked to the strain-induced magnetism and other magnetic phenomena already observed[10,11] in the bulk of SRO$_{214}$.

Another important implication of our results concerns the interplay between the normal-state TRSB phase, which we observe at the SRO$_{214}$ surface, and the TRSB existing in the superconducting state of SRO$_{214}$. Although the spin-triplet nature of the superconducting order parameter in SRO$_{214}$ is still under debate, previous studies show that TRSB due the pairing correlations with intrinsic chirality[7–9,45] (i.e., spin-singlet $d_{xz} \pm id_{yz}$ or spin-triplet $p_x \pm ip_y$) is a hallmark feature of the superconductivity in SRO$_{214}$. The normal-state TRSB due to orbital loop currents does not contradict the TRSB of the superconducting state in SRO$_{214}$, but it can bring further insights into the mechanism responsible for pairing formation with intrinsic chirality. This is because fluctuations of normal-state orbital loop currents, which are chiral in nature (they flow both clockwise and anticlockwise within the SRO$_{214}$ supercell), may extend from the SRO$_{214}$ surface, where such orbital loop currents get pinned and ordered, deeper into the bulk of SRO$_{214}$ and provide an unusual electronic mechanism responsible for the formation of Cooper pairs with intrinsic chirality in SRO$_{214}$.

A normal-state TRBS phase due to staggered orbital loop currents on the SRO$_{214}$ surface should in principle not favour the formation of a uniform superconducting phase in SRO$_{214}$ due to the incompatibility of the translation symmetry vectors of the two phases. Nevertheless, it is also possible that the superconducting order parameter in SRO$_{214}$ reconstructs and follows a non-uniform profile (e.g., a pair density wave profile) to accommodate the spatial variations of the orbital loop currents phase and minimize the magnetic fields associated with them, for example by driving pairing between the $d_{xy}$ and $(d_{xz}, d_{yz})$ orbitals. Both scenarios, meaning the suppression or the reconstruction of the superconducting order parameter due to its competition with the normal-state TRSB loop current phase, would lead to a superconducting state at the surface different from bulk of SRO$_{214}$— which can also account for some of the discrepancies between bulk-sensitive and surface-sensitive spectroscopy experiments reported to date on SRO$_{214}$.

Dipolar fields generated near edge dislocations, which are particularly relevant near the SRO$_{214}$ surface due to local strain inhomogeneities, can also be a source of time-reversal symmetry breaking[46] and therefore further contribute to the discrepancy in the symmetry of the superconducting order parameter determined based on the bulk- and surface-sensitive spectroscopy techniques.

It is interesting to observe that, regardless of the orbital loop current mechanism that we propose to explain the magnetism on the SRO$_{214}$ surface, this magnetism already represents a source of TRSB which can become more visible as superconductivity sets in, but without the TRSB being related to the superconducting order parameter per se. A normal-state TRSB phase can extend in principle to the entire sample as superconductivity sets in if an increase in the characteristic length scale of magnetism along the direction normal to the SRO$_{214}$ surface takes place. For this scenario to occur, the magnetic moment at the SRO$_{214}$ surface should generate dipolar fields in the superconducting state that are stronger than the critical field $H_{c1}$ of SRO$_{214}$. It has been reported that $H_{c1}$ is of ~10 G at $T = 0$ (ref. [47]), meaning that $H_{c1}$ is of the same strength as the dipolar fields probed by muons in our experiment. The dipolar fields that we detect by LE-μSR at the SRO$_{214}$ surface can therefore in principle induce the formation of a vortex liquid phase like that described in ref. [47]. This vortex liquid phase can give rise to a magnetic field distribution experienced by muons that is rather uniform, unlike the distribution corresponding to a vortex lattice, and possibly explain the TRSB in the superconducting state of SRO$_{214}$ reported in previous experiments based on bulk μSR[7,11].

Future studies will therefore have to clarify to which extent a normal-state TRSB phase at the surface can influence the superconducting state of SRO$_{214}$ and determine its symmetry and possible TRSB nature since superconducting pairing must occur in the presence of a phase breaking time-reversal symmetry already in the normal state. More generally, our results set a reference for the discovery of similar electronic phases in other compounds where orbital correlations play a role and suggest a novel mechanism originating in the normal state that can result in the formation of unconventional superconducting pairing associated with time-reversal symmetry breaking.

## Methods

**Sample preparation.** The Sr$_2$RuO$_4$ (SRO$_{214}$) single crystals used in this experiment are grown by the floating zone method and cleaved with a non-magnetic ZrO$_2$ to avoid contamination from magnetic impurities. The cleaved pieces are glued onto a Ni-coated aluminium plate to form a mosaic for the low-energy μSR measurements of size comparable to the muon beam (~2 cm in diameter).

**Electronic transport and structural characterization.** The electrical resistance of the SRO$_{214}$ crystals is measured in a four-probe configuration inside a cryogen-free system (Cryogenic Ltd.) with a base temperature of ~0.3 K using a current-biased setup with a current equal to or less than 0.1 mA.

High-angle X-ray diffraction measurements on the same SRO$_{214}$ samples are performed using a Panalytical X-Pert MRD PRO diffractometer. The diffractometer is equipped with a monochromatic CuK$_{\alpha1}$ radiation with wavelength $\lambda_{XRD} = 0.154\ 056$ nm obtained by a four-crystal Ge (220) asymmetric monochromator and a graded parabolic mirror positioned on the primary arm to the incident beam divergence to 0.12 arc sec. A triple-axis module with a triple-axis detector is used for the diffracted beam.

**Low-energy electron diffraction (LEED).** LEED measurements are carried out using a LEED spectrometer (SPECS) with electron energies of 185, 199, and 251 eV after in-situ cleaving the $SRO_{214}$ crystals at 10 K in an ultra-high vacuum with base pressure lower than $5 \times 10^{-11}$ Torr.

**Scanning SQUID.** SQUID measurements are done using a custom-built piezo-electric-based scanning SQUID microscope with a 1 μm pick-up loop[48] and magnetic sensitivity of $1\mu\Phi_0$ ($\Phi_0 = 2.0678 \times 10^{-15}$ T m$^2$ being the flux quantum). We use the scanning SQUID to image magnetic flux originating from the sample as a function of position. The magnetometry maps show the $z$-component of static magnetic flux near the surface of the sample. For the susceptometry measurements, we apply a local magnetic field of about 0.01–0.1 G using an on-chip field coil. Susceptometry is measured based on a standard lock-in technique and using the pick-up loop which resides at the centre of the field coil. An identical field coil surrounds a second pick-up loop used to correct for background magnetic fields in the magnetometry mode (gradiometric design).

## Data availability

The datasets generated during and/or analysed during the current study have been deposited in a Zenodo repository and they are publicly available at https://doi.org/10.5281/zenodo.5504281.

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

## Acknowledgements

A.D.B. and R.H. acknowledge funding from the Humboldt Foundation in the framework of a Sofja Kovalevskaja grant endowed by the Alexander von Humboldt foundation. A.D.B. also acknowledges funding from the Zukunftskolleg at the University of Konstanz. Y.M. acknowledges support from the JSPS (Nos. JP15H05852, JP15K21717, and JP17H06136) and, along with R.F., M.T.M., S.K., Y.Y., T.M., L.A.B.O.O., C.M.P.G., V.G., W.H., J.R., M.C., A.V., from the JSPS-EPSRC-CNR-IBS Core-to-core programme Oxide Superspin (Nos. EP/P026311/1 and JPJSCCA2017002). A.B. and B.K. acknowledge the European Research Council (Grant No. ERC-2019-COG-866236), the Israeli Science Foundation (Grant No. ISF-1251/19), the QuantERA ERA-NET (Project No. 731473). W.K. and C.K. acknowledge support from the Institute for Basic Science in Korea (Grant No. IBS-R009-G2). We also acknowledge A. Fittipaldi for support during the preparation of the experiment.

## Author contributions

A.D.B. conceived the idea of the project and supervised it. The experiment was designed by A.D.B., Z.S. and A.V. The samples were grown by R.F. and A.V., and characterized by

R.H., R.F., S.K., Y.Y., T.M., V.G. The LEED measurements were done by W.K. and C.K., and the SQUID measurements by A.B. and B.K. The muon measurements were performed by A.D.B., Z.S., R.H. with support from W.H., S.K., L.A.B.O.O., C.M.P.G., A.S., T.P., I.K. The muons results were analysed by A.D.B., Z.S., M.C. with inputs from A.V., Y.M., E.S., J.W.A.R. and other authors. The theoretical model was developed by M.T.M. and M.C. with support from A.R. and C.N. The paper was written by A.D.B. with help from M.C. and Z.S. and inputs and comments from all the other authors.

## Funding

## Competing interests

The authors declare no competing interests.
