## [Peer Review File · Nature Communications]

Reviewers' Comments:

Reviewer #1:

Remarks to the Author:

The manuscript of R. Fittipaldi et al. reports an important study of the surface magnetic properties of the unconventional superconductor Sr₂RuO₄. The nature of the superconducting state in Sr₂RuO₄ remains a mystery for nearly 30 years. Despite combined evidence for a broken time-reversal symmetry (BTRS) superconducting state in bulk, the convincing evidence for BTRS state at the surface is missing. The observation of unconventional weak magnetism at the surface, reported by R. Fittipaldi et al., provides a possible explanation for the discrepancy between bulk and surface sensitive probes. Therefore, I recommend the manuscript for publication in Nature Communications.

I have several questions regarding the interpretation of the data, which would be good to discuss in the manuscript.

1. The authors interpret the magnetic signal as arising due to loop currents producing nanoscale dipolar fields. This interpretation explains why the surface probes as a scanning SQUID microscopy could not resolve these fields. If I understand right, the fields should result in a very dense magnetic structure with the same magnetic field in every unit cell. In this case, one would expect to see a Gaussian muon spin relaxation at least in ZF- μ SR as an initial part of very slow muon spin oscillations (as expected from typical nuclear contribution). However, the relaxation process is exponential. Could authors explain this behaviour?

2. At the end of the manuscript, the authors discuss the interplay between the normal-state BTRS phase, which they observed at the surface, and the BTRS superconducting state existing in the superconducting state in bulk. In particular, they speculate that the mechanism responsible for pairing formation with intrinsic chirality may be related to fluctuating orbital loop currents in bulk. This assumption seems to be very interesting. However, at the surface, the static loop currents with a nanoscale period of \pm domains should be detrimental for BTRS superconductivity, where the characteristic length scales are of the order of the superconducting coherence length and penetration depth. Thus, I would expect a strong pair-breaking effect at the surface, and as a result, the superconducting state at the surface can be different from the bulk. This may be a possible reason for a discrepancy between surface probes and bulk sensitive experiments. Can authors discuss these aspects?

3. The proposal for the normal BTRS state with loop currents appeared first in a relation to cuprates. In particular, there were many efforts to search for such a state. However, despite some experimental claims Science 2011 332, 696-698, it seems the indication for this state was not observed by experimental techniques directly sensitive to magnetism such as μ SR. Therefore, the observation of this paper is relevant not only for the physics of Sr₂RuO₄ but also for the first time, providing a unique example (if the interpretation is correct) for the realization of this unusual quantum state of matter. Could the author discuss more the history of loop currents in relation to their results?

Reviewer #2:

Remarks to the Author:

Fittipaldi et al. report μ -SR experiments on the layered perovskite material Sr₂RuO₄ (SRO214). By varying the muon implantation energies they obtain a handle on the magnetic response as a function of the depth. An increase of the muon depolarization rate upon cooling indicative of a broadening of the distribution of local fields is taken as signature of enhanced magnetism. While seen for all used energies, the largest effect is seen for the smallest energy corresponding to the region closest to the surface. From this, the authors conclude upon magnetism at the surface of Sr₂RuO₄, which they try to reconcile with different origins in the Discussion. In particular, they put forward a model of orbital loop currents.

The finding of magnetism on the surface of SRO214 is interesting, but I do have a few questions and comments that require clarification:

(1) An onset temperature $T_{\text{on}} \sim 100$ K is identified, but I do not see any particular feature at 100 K neither in the shift of depolarization rates in Fig. 2 nor in Fig. 3b. Normalizing these data sets to different temperatures (i.e. 'vertically' shifting the data sets of different implantation energies with respect to each other) does not result in distinct behavior down to 50 K, given the scatter of the data points. The more pronounced enhancement of the low-energy data seems to occur primarily at temperatures below that. In the data shown in Fig. 2 it rather seems a general feature that the depolarization rate exhibits pronounced changes below 25 K for all energies, most pronounced for $E = 3$ keV.

(2) The local field (B_{loc}) variation mimics the temperature dependence of the local susceptibility obtained from NMR Knight shift. In the new μ -SR data a dependence on implantation energy is seen below 25 K, which the authors assign to a "strengthening of magnetism near the surface". In Ref. [11] cited in line 121, however, I do not find a particular change in Knight shift as a function of magnetic field. Can the authors approximate how large would be a magnetic field equivalent to such a change in Knight shift?

(3) The authors rule out impurities with the argument that the impurity density at the surface should not exceed the bulk in a randomly cleaved sample. The scanning SQUID measurements revealed small spots on the surface in Supplementary Fig. 3e,f. Irrespective of intrinsic or extrinsic origin, can such a surface structure give rise to a μ -SR signal similar to the observed one?

(4) On page 9 the authors further argue that spin textures with cancelling moments or spurious impurities with long-range interactions are ruled out by the monomodal $p(B_{\text{loc}})$ distribution. How would the local field distribution in Supplementary Fig. 5b look for randomly oriented, non-interacting moments on the surface? In case of overlapping contributions forming a broad peak, how could one distinguish this from a monomodal distribution?

(5) Regardless of the particular origin, the observation of magnetism at the surface is a very intriguing finding in view of the controversially discussed TRSB in the superconducting state of SRO214. I have a blunt question: the signatures of TRSB have been observed in measurement techniques that are sensitive to the surface of SRO214 (Kerr effect and μ -SR), but not in bulk probes. Can the surface magnetism be a potential source of TRSB, meaning that TRSB is unrelated to superconductivity but just becomes visible in the measurements as superconductivity sets in? The manuscript is well written and the results will appeal to a wide community interested in SRO214, hence it seems suitable for publication in Nature Communications, in principle. Yet, there are a few open questions remaining. I will be happy to review the manuscript again and provide a final recommendation once the authors have responded to the points above.

Reviewer #3:

Remarks to the Author:

Review of NCOMMS-21-12453-T

Unveiling unconventional magnetism at the surface of Sr₂RuO₄

This is an interesting and timely manuscript regarding a system of great current interest. However, there are significant problems with the data analysis, which makes the interpretation and conclusions potentially suspect. I recommend that the authors improve/show more of their data analysis to make a more convincing case for their conclusions. In its present form, the conclusions and interpretation are misleading. The paper should not be published without substantial improvements in presentation and analysis.

The main result is that muons implanted nearer to the surface of Sr₂RuO₄ see a somewhat larger increase in the transverse field muon spin relaxation rate with decreasing temperature below 300K (as shown in Figure 2). The authors have chosen to plot the change in relaxation rate, rather than absolute value of the relaxation rate, arguing that some systematic changes in the background signal make the total relaxation rate less reliable. The authors should perhaps show some of their raw data in a supplementary material section, and perhaps also a plot of the total relaxation rate, not just the change from the 300K value. The dashed lines (labelled as a guide to the eye) merely connect the highest and lowest temperature points and are essentially meaningless and should be removed.

There is no obvious transition temperature associated with the temperature dependence of the

relaxation rate, although the largest increase, especially in the higher energy (deeper implantation depth) data looks to occur around 25K. It would be useful to compare this low energy uSR measurement with more traditional bulk uSR (ideally on the same samples) to see if these effects are in fact visible far from the sample surface. This would have impact on the interpretation that these observations are due to the surfaces. It would also be useful to characterize a sample with no potential surface magnetism to see that there are no systematic effects involving the cryostat/apparatus at play.

The ZF results show weak relaxation, with a relaxation rate which is not given in the manuscript, but is clearly less than $0.1 \text{ microsec}^{-1}$. Instead of a straight line, the authors should actually fit the relaxing signal. This would correspond to a characteristic field of perhaps 1G, much less than the 10G estimate obtained from the decoupling field. This discrepancy would argue against the interpretation of static local fields and instead could indicate the presence of fluctuations. The early time data in the anti-parallel data (lower data set in Figure 4) appears distorted and should be omitted if this is the case.

Overall, the manuscript contains considerable modelling of the effects of orbital currents on the surface of Sr_2RuO_4 , but the underlying data and its interpretation are insufficient in their present form to justify such modelling.

We thank the referees for taking the time to review our manuscript comprehensively. We have addressed their comments, which we find very relevant and important, and we have modified the paper accordingly.

We provide a point-by-point response to the reviewers' comments and remarks below and explain the corresponding changes and additions that we have made to properly address them in the revised version of the manuscript.

Referee 1

The manuscript of R. Fittipaldi et al. reports an important study of the surface magnetic properties of the unconventional superconductor Sr₂RuO₄. The nature of the superconducting state in Sr₂RuO₄ remains a mystery for nearly 30 years. Despite combined evidence for a broken time-reversal symmetry (BTRS) superconducting state in bulk, the convincing evidence for BTRS state at the surface is missing. The observation of unconventional weak magnetism at the surface, reported by R. Fittipaldi et al., provides a possible explanation for the discrepancy between bulk and surface sensitive probes. Therefore, I recommend the manuscript for publication in Nature Communications.

We thank the referee for highlighting the importance of our work and in particular its relevance towards gaining a better understanding of the reason for the discrepancy existing about some experimental studies reported over the last 30 years on Sr₂RuO₄. We are glad that the referee *recommends the publication of our manuscript* in Nature Communications.

I have several questions regarding the interpretation of the data, which would be good to discuss in the manuscript.

1. The authors interpret the magnetic signal as arising due to loop currents producing nanoscale dipolar fields. This interpretation explains why the surface probes as a scanning SQUID microscopy could not resolve these fields. If I understand right, the fields should result in a very dense magnetic structure with the same magnetic field in every unit cell. In this case, one would expect to see a Gaussian muon spin relaxation at least in ZF- μ SR as an initial part of very slow muon spin oscillations (as expected from typical nuclear contribution). However, the relaxation process is exponential. Could authors explain this behaviour?

The referee is correct in their remark, as the muon spin relaxation is indeed Gaussian when all implanted muons experience the same field distribution in the sample, which is for example the case in bulk- μ SR. However, in low-energy μ SR (LE- μ SR) the muons' stopping depth distributions are relatively broad (see Fig. 1d), and the measured asymmetry at a given implantation energy E is a weighted average over a certain range of depths. Therefore, magnetic moments originating at the surface produce a varying dipolar field depending on the stopping depth. This results in a broadening of the measured field distribution, which in turn leads to better fits obtained using an exponential rather than a Gaussian distribution.

This point is now clarified in the revised manuscript **on page 4, line 68**, where we have added the following text:

We note that the finite width of the muons' implantation profiles in LE- μ SR (Fig. 1d) leads to a broadening of the field distribution experienced by the muons implanted at a given E . As a result, the asymmetry signal is better fitted assuming an exponential rather than a Gaussian relaxation rate, which is instead typically used in bulk- μ SR studies where all muons implanted in a homogeneous sample experience the same field distribution.

2. At the end of the manuscript, the authors discuss the interplay between the normal-state BTRS phase, which they observed at the surface, and the BTRS superconducting state existing in the superconducting state in bulk. In particular, they speculate that the mechanism responsible for pairing formation with intrinsic chirality may be related to fluctuating orbital loop currents in bulk. This assumption seems to

be very interesting. However, at the surface, the static loop currents with a nanoscale period of +- domains should be detrimental for BTRS superconductivity, where the characteristic length scales are of the order of the superconducting coherence length and penetration depth. Thus, I would expect a strong pair-breaking effect at the surface, and as a result, the superconducting state at the surface can be different from the bulk. This may be a possible reason for a discrepancy between surface probes and bulk sensitive experiments. Can authors discuss these aspects?

These are very insightful remarks, which give us the opportunity to further discuss the possible relations existing between the normal-state broken time reversal symmetry (BTRS) phase observed at the surface, which we report in the manuscript, and the superconducting BTRS phase existing in the bulk, as well as the correspondence between bulk and surface superconducting states. As pointed out by the referee, these aspects are of great relevance and timely in relation to the ongoing debate on the nature of the superconducting phase in Sr_2RuO_4 .

In the manuscript we speculate that the loop current state on the Sr_2RuO_4 surface might be linked to the time reversal symmetry breaking observed in the bulk of Sr_2RuO_4 below its superconducting phase transition. In this context, the remark made by the referee is very appropriate and correct, since the superconducting phase should nucleate differently depending on the presence of a static or fluctuating orbital loop current phase.

The staggered orbital loop current phase on the surface of Sr_2RuO_4 should in principle not coexist with a uniform superconducting phase due to the incompatibility of the translational symmetry vector. This is suggested also by previous studies on high temperature superconductors based on single or multi-band theory models, which suggest that the pseudogap state in high temperature superconductors emerges as result of the competition between a flux phase and a superconducting phase (see I. Affleck *et al.*, Phys. Rev. B **37**, 3774 (1988) or S. Chakravarty *et al.*, Phys. Rev. B **63**, 094503 (2001) or C. M. Varma, Phys. Rev. **73**, 155113 (2006)).

It must be observed, however, that the competition between a loop current phase with a non-uniform superconducting phase is in principle also possible, if the superconducting order parameter reconstructs in such a way that its spatial variation follows the variation of the staggered flux of the orbital loop currents (e.g., if the superconducting order parameter follows a pair density wave profile). For the specific case of Sr_2RuO_4 , there is an additional factor to take into account that is related to cooperation of the orbital degrees of freedom to minimize the strength of the time-reversal symmetry breaking field within each unit cell. The minimization of the magnetic field implies that electron pairing should also reduce pair-breaking effects within a single unit cell, for instance by favoring only inter-orbital pairing between d_{xy} and (d_{xz} , d_{yz}) orbitals.

Based on the above observations, in response to the referee's remark, we therefore conclude that two possible scenarios can occur as a result of the interplay between the normal-state BTRS phase and the superconducting BTRS phase: the coexistence of the orbital loop currents with superconductivity can be either detrimental for electron pairing or it can drive a reconstruction of the superconducting state to have a spatially non-uniform or orbital-selective order parameter.

On the other hand, in the bulk, due to the lack of loop current ordering, the resulting phase does not have the constraint mentioned for the surface. One can expect that orbital current fluctuations can mediate the electron pairing as we speculated in the manuscript. If a superconducting state with BTRS occurs, then, due to the presence of orbital loop excitations with non-trivial spatial form factor and the constraint of the tetragonal symmetry of the bulk crystal, a chiral $d+id$ state is a plausible candidate for the superconducting phase. For completeness, we mention that due to the spin-orbital entanglement of the electronic states, mixing with spin-triplet configurations are not excluded a priori.

Remarkably, both the suppression of the superconducting order parameter due to the staggered pair breaking potential or the pairing reconstruction within the distorted unit cell would lead to a surface configuration that is substantially different from that one in the bulk where the orbital loop phase is not the ground state in the normal state.

Such conclusions confirm the remark made by the referee that the presence of a static loop current phase on the surface would result in a superconducting phase with substantially different character in the bulk compared to the surface, which could possibly account for the observed discrepancies between surface and bulk probes.

This discussion is now included on page 13, line 329 of the revised manuscript where we have added the following text:

A normal-state TRBS phase due to staggered orbital loop currents on the SRO_{214} surface should in principle not favour the formation of a uniform superconducting phase in SRO_{214} due to the incompatibility of the translation symmetry vectors of the two phases. Nevertheless, it is also possible that the superconducting order parameter in SRO_{214} reconstructs and follows a non-uniform profile (e.g., a pair density wave profile) to accommodate the spatial variations of the orbital loop currents phase and minimize the magnetic fields associated with them, for example by driving pairing between the d_{xy} and (d_{xz} , d_{yz}) orbitals. Both scenarios, meaning the suppression or the reconstruction of the superconducting order parameter due to its competition with the normal-state TRSB loop current phase, would lead to a superconducting state at the surface different from bulk of SRO_{214} – which can also account for some of the discrepancies between bulk-sensitive and surface-sensitive spectroscopy experiments reported to date on SRO_{214} .

3. The proposal for the normal BTRS state with loop currents appeared first in a relation to cuprates. In particular, there were many efforts to search for such a state. However, despite some experimental claims Science 2011 332, 696-698, it seems the indication for this state was not observed by experimental techniques directly sensitive to magnetism such as μSR . Therefore, the observation of this paper is relevant not only for the physics of Sr_2RuO_4 but also for the first time, providing a unique example (if the interpretation is correct) for the realization of this unusual quantum state of matter. Could the author discuss more the history of loop currents in relation to their results?

We thank the referee for pointing out that our results are not only important because they contribute to a better understanding of the physics of Sr_2RuO_4 , but also because, as we claim in the manuscript, our results provide a concrete experimental evidence for the emergence of an orbital loop current state breaking time reversal symmetry in the normal state of a system like Sr_2RuO_4 , which several studies suggest it can break time reversal symmetry in the superconducting state. Our results therefore raise questions about the interplay existing between the loop current state and a time-reversal symmetry breaking state in Sr_2RuO_4 , and they also constitute a reference study for the detection of the orbital loop current state in other compounds similar to Sr_2RuO_4 .

As already reported in our manuscript, the existence of orbital loop currents had been already hypothesized in other compounds including cuprates and iron-based superconductors. For cuprates, it has been suggested that orbital loop currents with staggered flux can be the mechanism underlying intra-unit-cell antiferromagnetic ordering, which has been found in the pseudogap phase of the underdoped cuprates $\text{YBa}_2\text{Cu}_3\text{O}_{6+\delta}$ and $\text{HgBa}_2\text{CuO}_{4+\delta}$, using with spin-polarized neutron scattering (see B. Fauque *et al.*, Phys. Rev. Lett. **96**, 197001 (2006) or H. A. Mook *et al.*, Phys. Rev. B **78**, 020506 (R) (2008), Y. Li *et al.*, Nature **455**, 372 (2008), or Y. Li *et al.* Phys. Rev. B **84**, 224508 (2011)).

The original suggestion for the formation of an orbital loop current phase in cuprates was made by Varma (see C. M. Varma, Phys. Rev. Lett. **83**, 3538 (1999) or C. M. Varma, Phys. Rev. **73**, 155113 (2006)) and could explain the results obtained using spin-polarized neutrons. Other studies, however, have not confirmed the spin-polarized neutron results and therefore no conclusive evidence for the existence of an orbital loop current phase in cuprates has been demonstrated (W. H. P. Nielsen *et al.*, Phys. Rev. B **86**, 054510 (2012)). Suggestions on alternative experimental techniques that could be used to find alternative experimental signatures for the existence of an orbital-loop current phase in cuprates have also been made (see W. H. P. Nielsen *et al.*, Phys. Rev. B **86**, 054510 (2012) or S. Bulut *et al.*, Phys. Rev. B **92**, 195140 (2015)). Nevertheless, to the best of our knowledge, no work reporting the observation of such features has been published to date. Recently, it was also argued that the results obtained by polarized neutrons on underdoped $\text{YBa}_2\text{Cu}_3\text{O}_{6+\delta}$ cannot be explained on the basis of orbital loop currents like those proposed by Varma, as such orbital loop currents should lead to a magnetic moment which is at least one order of magnitude smaller ($\sim 0.01 \mu_B$) than that found experimentally by spin-polarized neutrons (see T. P. Croft *et al.*, Phys. Rev. B **96**, 214504 (2017))

The existence of an orbital loop current phases in cuprates remains controversial due to the fact that other competing phases such as, for example, charge density or spin density waves can also account for

the formation of a pseudogap in the normal state (see Z. Z. Li *et al.*, Physica C Supercond. **507**, 103 (2014) or B. Fauqué *et al.*, Phys. Rev. Lett. **96**, 197001 (2006)).

The orbital loop current state has also been theoretically predicted for iron-based superconductors, but never been observed experimentally. In particular, as suggested in the same paper reporting this theoretical result (see M. Klug *et al.*, Phys. Rev. B **97**, 155130 (2018)), the interplay between a spin density wave (SDW) existing in iron-based superconductors and a possible orbital loop current state can make it difficult to disentangle the two contributions and to find experimental signatures just for the orbital loop current state.

As reported in a very recent review paper (see P. Bourges *et al.*, <https://arxiv.org/abs/2103.13295>), the only two cases of systems where evidence for an orbital loop current phase has been claimed include the Mott insulator Sr_2IrO_4 (L. Zhao *et al.*, Nat. Phys. **12**, 36 (2016)) and the two-leg ladder cuprate $\text{Sr}_{14-x}\text{Ca}_x\text{Cu}_{24}\text{O}_{21}$ (D. Bounoua *et al.*, Commun. Phys. **3**, 123 (2020)). In the former case, signatures for an orbital loop current phase are found based on non-linear optical microscopy, and the authors suggest that this phase can be a possible path to realize superconductivity in Sr_2IrO_4 . No evidence for a time reversal symmetry breaking of this phase, however, has been found by other spectroscopy techniques like spin-polarized neutron scattering as reported in the same study (L. Zhao *et al.*, Nat. Phys. **12**, 36 (2016)). For $\text{Sr}_{14-x}\text{Ca}_x\text{Cu}_{24}\text{O}_{21}$, evidence for a phase breaking both time-reversal symmetry and parity has been reported using spin-polarized neutrons. The $\text{Sr}_{14-x}\text{Ca}_x\text{Cu}_{24}\text{O}_{21}$ system, however, is fundamentally different from Sr_2RuO_4 , which we consider in our study, as it is non-superconducting, and behaves like a spin liquid above a certain hole doping.

Further to the referee's suggestion, we have now added a short summary on the history of other studies (with corresponding references) on orbital loop current phases and discussed how such studies relate to our TRSB orbital loop current phase in Sr_2RuO_4 . This short summary is now reported on page **10**, line **234** of the manuscript, where we have added the following text:

Our theoretical analysis reported below shows that the origin of the normal-state TRSB phase probed on the surface of SRO_{214} can be ascribed to an orbital loop current with staggered magnetic flux. This orbital loop current phase is similar to that proposed to explain the intra-uni-cell antiferromagnetism in the pseudogap state of underdoped cuprates like $\text{YBa}_2\text{Cu}_3\text{O}_{6+\delta}$ and $\text{HgBa}_2\text{CuO}_{4+\delta}$ (refs.^{34,35}). The existence of an orbital loop current phase in cuprates, however, remains still controversial not only because earlier experimental evidence supporting the existence of this phase, and mostly based on spin-polarized neutrons^{34,35}, has not been confirmed by more recent studies, but also because alternative phases like charge density waves or spin density waves can equally account for the formation of the pseudogap in the normal state^{36,37}. Similarly, for materials like iron-based superconductors, it is difficult to demonstrate conclusive evidence for an orbital loop current phase based on experiments demonstrating evidence for TRSB in the normal state because of the simultaneous presence of a TRSB spin density wave in the same materials³⁸. To the best of our knowledge and as reported in ref.³⁹, the only two material systems for which evidence for an orbital loop current phase has been reported without effects that can be related to other coexisting TRSB phases include the Mott insulator SrIrO_4 using non-linear optical microscopy⁴⁰ and the two-leg ladder cuprate $\text{Sr}_{14-x}\text{Ca}_x\text{Cu}_{24}\text{O}_{21}$ using spin-polarized neutrons⁴¹. Spin-polarized neutron studies, however, have not confirmed the presence of normal-state TRSB orbital loop currents in Sr_2IrO_4 (ref.⁴⁰), whilst the orbital loop current phase reported for $\text{Sr}_{14-x}\text{Ca}_x\text{Cu}_{24}\text{O}_{21}$ cannot be directly correlated to that which we propose for SRO_{214} , since $\text{Sr}_{14-x}\text{Ca}_x\text{Cu}_{24}\text{O}_{21}$ has radically different physical properties in that it is non-superconducting and behaves like a spin liquid above a certain hole doping.

Referee 2

Fittipaldi et al. report mu-SR experiments on the layered perovskite material Sr₂RuO₄ (SRO₂₁₄). By varying the muon implantation energies they obtain a handle on the magnetic response as a function of the depth. An increase of the muon depolarization rate upon cooling indicative of a broadening of the distribution of local fields is taken as signature of enhanced magnetism. While seen for all used energies, the largest effect is seen for the smallest energy corresponding to the region closest to the surface. From this, the authors conclude upon magnetism at the surface of Sr₂RuO₄, which they try to reconcile with different origins in the Discussion. In particular, they put forward a model of orbital loop currents.

The finding of magnetism on the surface of SRO₂₁₄ is interesting, but I do have a few questions and comments that require clarification:

(1) An onset temperature $T_{\text{on}} \sim 100$ K is identified, but I do not see any particular feature at 100 K neither in the shift of depolarization rates in Fig. 2 nor in Fig. 3b. Normalizing these data sets to different temperatures (i.e. ‘vertically’ shifting the data sets of different implantation energies with respect to each other) does not result in distinct behavior down to 50 K, given the scatter of the data points. The more pronounced enhancement of the low-energy data seems to occur primarily at temperatures below that. In the data shown in Fig. 2 it rather seems a general feature that the depolarization rate exhibits pronounced changes below 25 K for all energies, most pronounced for $E = 3$ keV.

We thank the referee for this remark. We note that a similar comment was also made by referee 3, for which we also refer to the point 2 of our reply to referee 3 (reported below).

To address the referee’s comment, we have had a closer look into the raw asymmetry data, and concluded that indeed T_{on} is lower than 100 K, as noted by the referee, but not lower than 50 K. For the data set at $E = 3$ keV, our new analysis of the raw asymmetry data shows that the increase in the muon depolarization rate λ with decreasing temperature (T) changes slope between 50 K and 75 K, thus suggesting that the onset temperature of magnetism T_{on} lies within this temperature range. To further support our claim, in the revised version of the manuscript, we have now added a new **Supplementary Figure 7** showing raw asymmetry data and the corresponding fits for a few representative T s at the three E s (i.e., 3 keV, 6 keV and 14 keV) shown in Fig. 2. Based on the results of this analysis, we now state that T_{on} lies between 50 K and 75 K in the revised manuscript.

Although we can establish that $50 \text{ K} < T_{\text{on}} < 75 \text{ K}$ from the analysis of raw asymmetry data at $E = 3$ keV, we agree with the referee that the data sets collected at higher energies (i.e., $E = 6$ keV and $E = 14$ keV) in Fig. 2 suggest that the depolarization rate increases at a T lower than 50 K. The raw asymmetry profiles (see also response to referee 3) indeed show that, unlike for $E = 3$ keV where a significant change in the damping of the asymmetry occurs at $T = 50$ K compared to higher T s, the damping does not change significantly until a much lower T (lower than 25 K) is reached.

These results, however, do not contradict the above analysis and the value of T_{on} estimated from the data set at $E = 3$ keV, since the magnetism that we observe originates from the surface of SRO₂₁₄, as also proven by the measurement data reported in Fig. 3 of the paper. The surface nature of the magnetism implies that the onset temperature determined from the muons’ depolarization rate should be higher on the surface compared to bulk. In other terms, since the magnetism originates at the surface of SRO₂₁₄, the implanted muons should only experience an increase in their depolarization rate deeper inside SRO₂₁₄ (i.e., at higher E s) when the magnetism on the surface has become sufficiently strong, which occurs when T has been decreased well below the onset of the magnetic phase transition at T_{on} .

Further to this new analysis which we have carried out to address the referees’ accurate remarks, we have corrected the previous estimated value of $T_{\text{on}} \sim 100$ K and in the revised version of the manuscript we now claim that $50 \text{ K} < T_{\text{on}} < 75 \text{ K}$. We have also added the following text on **page 5 line 94** of the revised manuscript to clarify that the onset temperature measured for magnetism should decrease at higher implantation E s:

The $\Delta\lambda$ values reported in Fig. 2, in combination with the corresponding raw asymmetry profiles with corresponding fits reported in the Supplementary Information, show that $\Delta\lambda$ at $E = 3$ keV significantly changes slope at a T between 50 K and 75 K, which we identify as the T_{on} of the magnetism. The data sets in Fig. 2 and Supplementary Fig. 7 for $E = 6$ keV and $E = 14$ keV also demonstrate that the onset temperature of the magnetism detected by muons decreases at higher implantation depths, since $\Delta\lambda$ for $E \geq 6$ keV does not change significantly until a $T \sim 25$ K is reached, which is lower than the estimated $50 \text{ K} < T_{on} < 75 \text{ K}$. This result further confirms the surface nature of the magnetism that we measure in SRO_{214} because the muons implanted deeper inside SRO_{214} only experience an increase in their depolarization rate when the magnetism on the surface has become sufficiently strong, which occurs when T has been decreased well below the onset of the magnetic phase transition at T_{on} .

In addition, we have modified Fig. 2 in the revised version of the manuscript, also to address the comments made by referee 3 (see point 2 of the response to referee 3 below), as shown below.

Fig. 1: Temperature dependence of magnetism in SRO_{214} at different implantation depths.

Shift in muon depolarization rate, $\Delta\lambda$, from the λ value measured at $T = 270$ K as a function of temperature T measured in a TF setup (inset) with $B_{ext} = 100$ Gauss at different implantation energy E values. The solid grey line serves as guide to the eye and marks the T range (grey shaded region) where $\Delta\lambda$ changes slope for $E = 3$ keV, which we identify as the onset temperature T_{on} of the magnetism in SRO_{214} .

Last, as discussed above, we have added a new **Supplementary Figure 7** to the manuscript showing a few representative asymmetry profiles measured at different energies, from which we estimate that T_{on} is between 50 K and 75 K. This new **Supplementary Figure 7** is shown in our response to referee 3 reported below (see point 2 of the reply to referee 3).

(2) The local field (B_{loc}) variation mimics the temperature dependence of the local susceptibility obtained from NMR Knight shift. In the new mu-SR data a dependence on implantation energy is seen below 25 K, which the authors assign to a “strengthening of magnetism near the surface”. In Ref. [11] cited in line 121, however, I do not find a particular change in Knight shift as a function of magnetic field. Can the authors approximate how large would be a magnetic field equivalent to such a change in Knight shift?

The referee is correct that in ref. [11] they do not study the change in the NMR Knight shift as a function of the magnetic field. In our LE- μ SR experiment, the relative change in B_{loc} compared to the applied field experienced by muons, meaning $(B_{loc} - B_{ext})/B_{ext}$, is proportional to the sample’s susceptibility and directly proportional to the Knight shift, as measured in ref. [11] ((T. Imai *et al.*, Phys. Rev. Lett. **81**, 3006 (1998)). We would like here to clarify the following points to better explain the

connection that we intended to draw between our LE- μ SR results and the NMR Knight shift measurements is reported in ref. [11]:

- 1) Independently on the energy (i.e., both at $E = 3$ keV and 14 keV), when $B_{\text{ext}} = 1500$ Gauss we observe a relative change in B_{loc} compared to B_{ext} reaching a maximum of about 0.2 Gauss/1500 Gauss $\sim 0.01\%$ at $T \sim 25$ K. The observed temperature dependence is similar to that observed in ref. [11] for the susceptibility χ_{xy} extracted from ^{17}O -NMR measurements. An accurate comparison between our LE- μ SR and the NMR measurements, however, it is unfortunately not possible since we do not know the strength and the nature of the coupling between implanted muons and SRO_{214} .
- 2) We did not state in the manuscript that we observe a Knight shift also at a lower applied field $B_{\text{ext}} = 100$ Gauss. The reason for this is because we do not have the resolution to measure an equivalent Knight shift to that measured with higher B_{ext} , which should be the 0.01% of 100 Gauss – which would correspond to a magnetic field of 0.01 Gauss that is well beyond the resolution of the LE- μ SR technique. We discuss this point in the original version of the manuscript on page 6, line 132.
- 3) When we argue a “strengthening of magnetism near the surface”, we refer to the shift in B_{loc} that we observe in $B_{\text{ext}} = 1500$ Gauss when E is decreased from 14 keV to 3 keV (see Fig. 3a of the manuscript). Such increase corresponds to an increase in susceptibility (or equivalently in the Knight shift) at the SRO_{214} surface. This result, however, which we observe by LE- μ SR, does not have an equivalent in ref. [11], since the ^{17}O NMR measurements in ref. [11] are done on the bulk of SRO_{214} meaning that they do not have a corresponding equivalent to our LE- μ SR measurements on the surface of SRO_{214} at $E = 3$ keV.

We hope these points help clarify the original statements made in the manuscript, thus addressing the question made by the referee.

(3) The authors rule out impurities with the argument that the impurity density at the surface should not exceed the bulk in a randomly cleaved sample. The scanning SQUID measurements revealed small spots on the surface in Supplementary Fig. 3e,f. Irrespective of intrinsic or extrinsic origin, can such a surface structure give rise to a μ -SR signal similar to the observed one?

The relative surface area covered by such impurities is very small (much less than 1%) and their contribution to the μ SR signal will be correspondingly small. In other terms, the enhancement in the muons' depolarization rate, which is observed for the entire signal, cannot be accounted for by such impurities which will affect only a small fraction of the implanted muons.

We now clarify this point on page 9, line 194 of the revised manuscript where we have added the following text:

We note that, in our scanning SQUID measurements, we can only detect small magnetic spots on the SRO_{214} crystals (Supplementary Fig. 3), most likely of extrinsic origin and possibly introduced during the cleaving process. The magnetic spots, however, only occupy a very small area of the sample (less than 1%) and therefore they would only affect a small fraction of the implanted muons, meaning that they cannot account for the uniform increase in the depolarization rate measured in the LE- μ SR signal below T_{on} .

(4) On page 9 the authors further argue that spin textures with cancelling moments or spurious impurities with long-range interactions are ruled out by the monomodal $p(B_{\text{loc}})$ distribution. How would the local field distribution in Supplementary Fig. 5b look for randomly oriented, non-interacting moments on the surface? In case of overlapping contributions forming a broad peak, how could one distinguish this from a monomodal distribution?

We agree with the referee, and this was an oversight from our side, since spin textures with cancelling moments may indeed result in a monomodal distribution in the case of overlapping contributions from the spin sublattices forming a broad peak. We can still, however, exclude these spin textures as the origin for the magnetism which we detect in SRO_{214} on the basis of the same argument that we use to

rule out correlations between magnetic impurities embedded inside a metallic Fermi sea. This is because magnetic spin textures with an onset temperature T_{on} as high as that which we probe in SRO₂₁₄ (i.e., $50 \text{ K} < T_{\text{on}} < 75 \text{ K}$) would require magnetic moments much larger than the values which we measure ($\ll 0.01 \mu_{\text{B}}/\text{Ru atom}$) to order. Also, our scanning SQUID measurements also confirm that the samples do not contain any (dense) magnetic moments that can order at low temperatures.

Further to this referee's remark and based on the above considerations, we have therefore corrected our argument to rule out spin textures with cancelling magnetic moments on **page 9, line 203** of the manuscript, where the manuscript now reads as follows:

Further to conventional ferromagnetism, we also rule out magnetism due to spin textures with cancelling moments^{24,25} or to correlations between spurious magnetic impurities as possible explanation for our results in SRO₂₁₄. This is because the appearance of such magnetic phases due to long-ranged correlations between magnetic spins or magnetic impurities embedded into a metallic Fermi sea at the relatively high onset temperature $50 \text{ K} < T_{\text{on}} < 75 \text{ K}$ we measure would require a large strength of the Ruderman-Kittel-Kasuya-Yosida interaction and/or a strong crystal field anisotropy. This should, however, result in a magnetic moment much larger than the value that we measure (much less than $0.01 \mu_{\text{B}}/\text{Ru atom}$).

(5) Regardless of the particular origin, the observation of magnetism at the surface is a very intriguing finding in view of the controversially discussed TRSB in the superconducting state of SRO₂₁₄. I have a blunt question: the signatures of TRSB have been observed in measurement techniques that are sensitive to the surface of SRO₂₁₄ (Kerr effect and $\mu\text{-SR}$), but not in bulk probes. Can the surface magnetism be a potential source of TRSB, meaning that TRSB is unrelated to superconductivity but just becomes visible in the measurements as superconductivity sets in?

We agree with the referee that the relation between the surface magnetism which we observe and the TRSB related to the superconducting phase of SRO₂₁₄ can be indeed important to solve the puzzle on the nature of the superconductivity in SRO₂₁₄ and its TRSB nature.

The possibility described by the referee, meaning that surface magnetism gets amplified when superconductivity sets in, represents indeed a likely scenario. Although low-energy muon spectroscopy (LE- μSR) is probably the only surface technique that can address this question since it can probe surface states both with nanometer-depth resolution and with the very high sensitivity needed to resolve the extremely weak magnetic moments associated with orbital loop currents ($< 0.01 \mu_{\text{B}}/\text{Ru atom}$), this technique cannot be used currently to study the evolution of the surface magnetic states below the superconducting transition of SRO₂₁₄. This is because the base temperature of the LE- μSR setup is of $\sim 2.3 \text{ K}$, which is above the superconducting critical temperature T_{c} of SRO₂₁₄ of $\sim 1.5 \text{ K}$.

Whilst future studies are needed to address the question raised by the referee, we discuss possible scenarios that may explain how the TRSB due to surface magnetism can become more visible as superconductivity sets in, but without being related to nature of the superconducting order parameter per se. First, we note that for the normal-state TRSB to extend to the entire sample as superconductivity sets in, a significant increase in the characteristic length scale of the magnetism along the direction normal to the sample surface is needed. For it to occur, this scenario would require a magnetic moment at the surface generating dipolar fields stronger than then the critical field H_{c1} of SRO₂₁₄ – which is not of $\sim 10 \text{ Gauss}$ at $T = 0$ and therefore comparable dipolar fields probed by muons in our experiment (see D. Shibata *et al.*, Phys. Rev. B **91**, 104514 (2015)). These dipolar fields can induce the formation of a vortex liquid phase (see also D. Shibata *et al.*, Phys. Rev. B **91**, 104514 (2015)), which would give rise to a magnetic field distribution experienced by muons that is rather uniform, unlike the distribution corresponding to a vortex lattice, and possibly explain the TRSB reported in experiments on SRO₂₁₄ based on bulk μSR .

An alternative scenario to the one described refers to the possibility for wave fluctuations of the orbital loop current state to act the pairing glue for superconducting correlations in the bulk (see also point 2 of the reply to the Referee 1). In this case, the intrinsic TRSB character of the loop current state would naturally favor the formation of an electron pairing that breaks time reversal symmetry. Furthermore, the form factor of the loop current state, which does not have a s -wave nature, would

result into a non-zero angular momentum of the Cooper pair wave function. We argue that a chiral $d+id$ superconducting state is the most likely candidate to get stabilized in this framework.

As a follow up of the discussion presented at the point 2 of the reply to the Referee 1 (to which we refer for further details), we would also like to state here that, independently on the interplay between the loop current phase and superconductivity, the presence of a static loop current magnetic phase on the surface of SRO_{214} can result in a surface superconducting phase with a substantially different character compared to the bulk superconducting phase, which provides a possible physical scenario for the observed discrepancies between surface and bulk probes.

Further to this insightful observation made by referee, we now discuss in the revised manuscript how a TRSB phase in the normal state can be a potential source of TRSB in the superconducting state, but without being related to superconductivity per se. To this purpose, we have added the following text on **page 14 line 340** of the revised manuscript:

It is interesting to observe that, regardless of the orbital loop current mechanism that we propose to explain the magnetism on the SRO_{214} surface, this magnetism already represents a source of TRSB which can become more visible as superconductivity sets in, but without the TRSB being related to the superconducting order parameter per se. A normal-state TRSB phase can extend in principle to the entire sample as superconductivity sets in, if an increase in the characteristic length scale of magnetism along the direction normal to the SRO_{214} surface takes place. For this scenario to occur, the magnetic moment at the SRO_{214} surface should generate dipolar fields in the superconducting state that are stronger than the critical field H_{c1} of SRO_{214} . It has been reported that H_{c1} is of ~ 10 Gauss at $T = 0$ (ref. ⁴⁴), meaning that H_{c1} is of the same strength as the dipolar fields probed by muons in our experiment. The dipolar fields that we detect by LE- μSR at the SRO_{214} surface can therefore in principle induce the formation of a vortex liquid phase like that described in ref.⁴⁴. This vortex liquid phase can give rise to a magnetic field distribution experienced by muons that is rather uniform, unlike the distribution corresponding to a vortex lattice, and possibly explain the TRSB in the superconducting state of SRO_{214} reported in previous experiments based on bulk μSR ^{8,12}.

The manuscript is well written and the results will appeal to a wide community interested in SRO_{214} , hence it seems suitable for publication in Nature Communications, in principle. Yet, there are a few open questions remaining. I will be happy to review the manuscript again and provide a final recommendation once the authors have responded to the points above.

We thank for the referee for finding our manuscript *suitable for publication in Nature Communications*. We have carefully addressed all the referee's remarks, which has certainly contributed to improving the quality of our manuscript, and we believe that the referee will find our answers satisfactorily.

Referee 3

This is an interesting and timely manuscript regarding a system of great current interest. However, there are significant problems with the data analysis, which makes the interpretation and conclusions potentially suspect. I recommend that the authors improve/show more of their data analysis to make a more convincing case for their conclusions. In its present form, the conclusions and interpretation are misleading. The paper should not be published without substantial improvements in presentation and analysis.

We thank the referee for finding our manuscript *interesting and timely*. We have taken the referee's comments into account and done an additional analysis of our data to further validate our claims. This new analysis, which has been added to the manuscript, has certainly contributed to strengthening our claims and to further validate the interpretation of our results. We are confident that we have properly addressed all the remarks made by the referee, as explained in our point-by-point response below, and we therefore believe that our revised manuscript is now suitable for publication in *Nature Communication*.

(1) The main result is that muons implanted nearer to the surface of Sr₂RuO₄ see a somewhat larger increase in the transverse field muon spin relaxation rate with decreasing temperature below 300K (as shown in Figure 2). The authors have chosen to plot the change in relaxation rate, rather than absolute value of the relaxation rate, arguing that some systematic changes in the background signal make the total relaxation rate less reliable. The authors should perhaps show some of their raw data in a supplementary material section, and perhaps also a plot of the total relaxation rate, not just the change from the 300K value. The dashed lines (labelled as a guide to the eye) merely connect the highest and lowest temperature points and are essentially meaningless and should be removed.

In the manuscript we have decided to plot the change in relaxation rate other than the absolute value of the relation rate because the number of backscattered muons increases dramatically with decreasing implantation energy E . It is generally known that when E is below ~ 5 keV, the backscattered muons produce a systematic and unphysical increase in the relaxation rate even if implanted in a non-magnetic material such Ag or Au. Therefore, direct comparison between the relaxation rates extracted from the raw data may be misleading, since the large differences in the absolute values of the relaxation rates at low E (e.g., $E = 3$ keV) and higher E s do not reflect the actual changes in the physical properties of the sample itself, but they are simply due to a variation in the number of backscattered muons. On the other hand, any change as a function of temperature must be due to variations in the physical properties of the sample. This is the reason behind our choice to plot the increase in damping rate other than its absolute value.

Further to the referee's remark, nonetheless we have decided to also show the temperature evolution of the absolute relaxation rate at different E s, which has been added now to the revised manuscript as Supplementary Fig. 6 and it is reported for completeness also below. The data in Supplementary Fig. 6 clearly show that the damping rate at $E = 3$ keV increases by more than three times when temperature is decreased from 270 K to 5 K.

We discuss now the points above in the Supplementary Information of the manuscript on **page 7, line 158**, where we have included, in addition to the **Supplementary Figure 6** (shown below), also the following text:

we note that in Fig. 2 of the manuscript we do not report the T -dependence of λ but the T -dependence of the shift in the depolarization rate, $\Delta\lambda(T)$, determined from the λ value measured at $T = 270$ K. The reason for our choice to show the $\Delta\lambda(T)$ profiles at different E s in Fig. 2 of the manuscript other than the $\lambda(T)$ profiles is because the large differences in the absolute values of λ at low E (e.g., $E = 3$ keV) compared to the of λ values at higher E s (e.g., $E = 6$ keV and 14 keV) do not reflect actual changes in the physical properties of the SRO₂₁₄ samples, but they are simply due to a variation in the number of backscattered muons. The $\lambda(T)$ profiles measured at different E s are reported for completeness in Supplementary Fig. 6. It is worth noting that the data in Supplementary Fig. 6 show that λ increases by a factor larger than 3 at $E = 3$ keV when T is decreased from 270 K down to 5 K.

Supplementary Figure 6: Temperature dependence of the depolarization rate in SRO₂₁₄ at different muons' implantation depths.

Depolarization rate λ as a function of temperature T measured in a TF setup (inset) with an applied magnetic field $B_{\text{ext}} = 100$ Gauss at different implantation energy E values: 3 keV (red symbols), 6 keV (orange symbols) and 14 keV (blue symbols).

In addition, we have revised Fig. 2 of the manuscript and removed the dashed lines, which we agree they did not follow the trend of the experimental points. The revised Fig. 2 is also reported above in our reply to point 1 of referee 2.

(2) *There is no obvious transition temperature associated with the temperature dependence of the relaxation rate, although the largest increase, especially in the higher energy (deeper implantation depth) data looks to occur around 25K.*

As requested by the referee, we have now added raw asymmetry data and their corresponding fits to the manuscript in a new Supplementary Figure 7. Each panel of this new figure, which is reported for clarity also below, shows the asymmetry measured at a different energy ($E = 3$ keV, 6 keV and 14 keV) for three different representative temperature values ($T = 5$ K, 50 K and 270 K). The temperatures are chosen to show how the damping in the asymmetry signal (i.e., the muon depolarization rate λ) changes across the temperature onset, T_{on} , of the magnetism at the Sr₂RuO₄ surface. The new analysis discussed below, along with the other asymmetry curves at $T = 75$ K and $T = 100$ K which we have analyzed (but not reported in the figure below to be able to appreciate small variations in the three asymmetry profiles already shown), supports our claim that T_{on} lies between 50 K and 75 K.

We note in fact that the asymmetry signal at $E = 3$ keV (panel (a) in the Figure below) exhibits a significant increase in damping from the value measured at 270 K (black curve) already for T equal to 50 K (light blue curve). This is evidenced by the fact that at $E = 3$ keV the asymmetry measured at $T = 50$ K already deviates at $t \sim 1.5$ μs from the asymmetry profile at $T = 270$ K. The data in the panels (b) and (c) of the same Figure, however, also suggest that at higher energies, namely at $E = 6$ keV and $E = 14$ keV, the increase in damping as T is decreased from 270 K to 50 K is not as significant as at $E = 3$ keV, and a very small separation between the asymmetry curves at $T = 50$ K and $T = 270$ K only becomes visible for a relaxation time larger than 4 μs .

The raw asymmetry data therefore suggest, consistently with the increase in λ shift, $\Delta\lambda$, shown in Fig. 2 of the manuscript, that closer to the Sr₂RuO₄ surface at $E = 3$ keV magnetism – which is associated with an increase in the slope of $\Delta\lambda$ – sets in at a temperature $50 \text{ K} < T_{\text{on}} < 75 \text{ K}$.

Also, we note that this magnetism can only be probed at higher energies when it becomes stronger on the Sr₂RuO₄ surface as T is further reduced below 50 K. This can be inferred from the fact that there is a separation in the asymmetry profiles measured at $T = 50$ K and $T = 5$ K for $E = 6$ keV (panel (b) in

the Figure below) due to an increase in damping, but this separation between the curves measured at the same temperatures is hardly visible at $E = 14$ keV.

We discuss now all these points in the Supplementary Information, where we have added the following text on page 8, line 177:

As explained in the manuscript, from the T -dependence of $\Delta\lambda(T)$ profiles obtained in TF, we also estimate that $50 \text{ K} < T_{on} < 75 \text{ K}$. This result, which we infer based on the $\Delta\lambda(T)$ profiles shown in Fig. 2 of the manuscript, is also evidenced by the data sets in Supplementary Fig. 7, where we show the raw asymmetry data and corresponding fits measured at a few representative temperatures ($T = 5 \text{ K}$, 50 K and 270 K) at three different energies ($E = 3 \text{ keV}$, 6 keV and 14 keV). In particular, Supplementary Fig. 7a shows that the asymmetry signal at $E = 3 \text{ keV}$ (exhibits a significant increase in muons' depolarization from the value measured at 270 K (black curve) already for $T = 50 \text{ K}$ (light blue curve), as evidenced by the fact that the asymmetry curve at $T = 50 \text{ K}$ already deviates at $t \sim 1.5 \mu\text{s}$ from the asymmetry profile at $T = 270 \text{ K}$. This is in contrast with the data reported in Supplementary Fig. 7b, c (showing the data for $E = 6 \text{ keV}$ and 14 keV) where a very small separation between the asymmetry curves at $T = 50 \text{ K}$ and $T = 270 \text{ K}$ only becomes visible for a relaxation time larger than $4 \mu\text{s}$.

The raw asymmetry data therefore suggest, consistently with the $\Delta\lambda$ values extracted from these asymmetry curves and reported for more T values in Fig. 2 of the manuscript, that closer to the SRO_{214} surface at $E = 3 \text{ keV}$, the magnetism – which is associated with an increase in the slope of $\Delta\lambda$ – sets in at an onset temperature $50 \text{ K} < T_{on} < 75 \text{ K}$.

As mentioned above, we report raw asymmetry data and corresponding fits for a few representative temperatures and energies in the Supplementary Figure 7, which is shown for clarity also below.

Further modifications made to the manuscript to address related questions made by referee 2 are also reported above in our reply to point 1 of referee 2.

(3) It would be useful to compare this low energy μ SR measurement with more traditional bulk μ SR (ideally on the same samples) to see if these effects are in fact visible far from the sample surface. This would have impact on the interpretation that these observations are due to the surfaces. It would also be useful to characterize a sample with no potential surface magnetism to see that there are no systematic effects involving the cryostat/apparatus at play.

As already noted in the original version of the manuscript on page 6 line 107, we have collected the muon data sets reported in the main paper on two different batches of SRO₂₁₄ samples over three beam time sessions, during which we used different cryostats and different magnets. All the sets of data measured show the same enhancement in the magnetism near the SRO₂₁₄ surface, which rules out any systematic effects due to the cryostat/apparatus or to the specific batch of samples investigated.

We also point out that our best “bulk” reference signal is obtained from LE- μ SR measurements performed on the same SRO₂₁₄ samples at higher implantation energies, which show a much weaker temperature dependence compared to the surface measurements performed at lower energies on these samples (see Figs. 2 and 3 of the manuscript). We do not expect to get a better reference from bulk μ SR measurements on these SRO₂₁₄ samples, since we think we already have the ideal reference. Moreover, bulk μ SR data on clean SRO₂₁₄ samples with $T_c \sim 1.5$ K (like our samples) are already available in the literature and show no strong temperature dependence of the damping rate in the normal state (see, e.g., G. M. Luke *et al.*, Physica B: Condens. Matter. **289**, 373 (2000)).

Nevertheless, although we think we have already carried the best “bulk” reference measurements, we have also performed a few bulk- μ SR measurements on SRO₂₁₄ single crystals from the same batch reported in our study, as asked by the referee. These measurements confirm our statements above and show that λ shows no significant variation between room temperature and 5 K both in an applied field $B_{\text{ext}} = 100$ Gauss and also in $B_{\text{ext}} = 1500$ Gauss.

We hope that these additional tests, which we have performed using bulk- μ SR other than LE- μ SR, and the considerations above regarding the LE- μ SR data at $E = 14$ keV already reported in the manuscript address the referee’s concerns.

(4) The ZF results show weak relaxation, with a relaxation rate which is not given in the manuscript but is clearly less than $0.1 \text{ microsec}^{-1}$. Instead of a straight line, the authors should actually fit the relaxing signal. This would correspond to a characteristic field of perhaps 1G, much less than the 10G estimate obtained from the decoupling field. This discrepancy would argue against the interpretation of static local fields and instead could indicate the presence of fluctuations. The early time data in the anti-parallel data (lower data set in Figure 4) appears distorted and should be omitted if this is the case.

The referee is correct in their assessment, as the relaxation rate is indeed very small. We should point out here that:

- 1- As the referee is aware, the data reported in the manuscript are collected by LE- μ SR spectroscopy, which implies that the data presented are based on lower statistics due to the limited time and lower muons rate compared to bulk- μ SR experiments.
- 2- In LE- μ SR measurements in the longitudinal field (LF) configuration, the maximum asymmetry is quite small in intensity (< 0.1) due to the geometry of the LE- μ SR spectrometer.
- 3- The damping rate measured in SRO₂₁₄ is quite small and of the same level as the damping rate just due to the background in the LF/ZF setup (the backscattered muons and tail of the symmetry in the Ni backing plate both decay with a rate also of $\sim 0.1 \text{ } \mu\text{s}^{-1}$).
- 4- In our ZF/LF measurement we see no evidence for two signals, meaning for a fast and slow relaxing component in the asymmetry.

Based on the above considerations, it is clear that any attempt to fit the LF/ZF asymmetry data according to a two-component model will result in an over-parametrization and therefore in meaningless fits in

our study. Our conservative evaluation of the damping rate, nonetheless, based on our analysis still sets a lower limit for estimating the strength of the local magnetic fields in SRO₂₁₄.

Last, we would like to point out that, if there were two components in the LF/ZF data, these should become even more visible in the TF measurements, which is clearly not the case in our experiment, as shown by the asymmetry profiles reported in the Supplementary Information.

In response to the second remark made by the referee on the early-time data, we agree that there is indeed a distortion in the anti-parallel data at early times. However, it is important to note here that the data of the two opposite polarizations is fitted with a common polarization function (same damping rate but opposite initial polarization). Therefore, the parameters obtained from the fits are highly constrained and they take into consideration the data from both polarization directions, which is crucial in this case due to the low signal-to-noise ratio of the measurements (due to the low asymmetry intensity and low statistics). We have also followed the referee's suggestion and omitted the distorted time range in our current fits and in the revised manuscript.

We have had a closer look at our analysis of the LF/ZF asymmetry data and repeated the fits with the theoretical function expected for a Lorentzian static field distribution in ZF and LF (i.e. fit to a static exponential/Lorentzian Kubo-Toyabe function in ZF and LF; see Y. J. Uemura *et al.*, Phys. Rev. B **31**, 546 (1985)), assuming that the local static fields do not change. In this fit, there is only one free physical parameter which is the Lorentzian field distribution width, meaning that the field dependence (decoupling) is determined by the theoretical function for the corresponding applied field. The fit gives a value of the half width at half maximum (HWHM) of the field distribution of ~ 0.5 Gauss which is consistent with the value that we estimate in the original manuscript for the local static fields probed by muons near the surface of SRO₂₁₄.

The results of this new analysis are reported in the figure below, which represents Figure 4 of the revised manuscript. We note that the new fits in the figure below do not include the distortion at early times, which addresses the remark made by the referee.

Overall, the manuscript contains considerable modelling of the effects of orbital currents on the surface of Sr₂RuO₄, but the underlying data and its interpretation are insufficient in their present form to justify such modelling.

We respectfully disagree with the referee on this point. The measured effect can be summarized as follows:

- The damping rate of the muon spin polarization increases with decreasing temperature near the surface of the SRO₂₁₄ crystals. This is in contrast with the weaker temperature dependence observed deeper in the bulk of SRO₂₁₄.
- We establish that the origin of the measured damping is weak local static magnetic fields.
- The onset temperature for the appearance of these static fields is relatively high and between 50 K and 75 K.

These are experimental observations that are significant even when considering any possible systematic effects due to the LE- μ SR apparatus. In addition, systematic effects are directly ruled out by us through verifying the reproducibility of the results on different batches of SRO₂₁₄ samples and using different cryostats and different magnets for the collection of the measurement data.

As we clarify above, different scenarios that could be depicted to explain our experimental observations on the basis of common magnetic ordering behavior, such as magnetic impurities (interacting or non-interacting), can be easily ruled out since they are not consistent with the experimental data. On the other hand, the theoretical model brought forward in the manuscript can explain our observations fully in a very simple yet elegant way and can also link our results to several other puzzling findings reported in the literature for the bulk of SRO₂₁₄.

We have carefully addressed all the remarks made by the referees, collected additional bulk measurements and revised our data analysis, which has certainly contributed to improving the quality of the manuscript and to confirming the veracity of our conclusions. On the basis of this and the above considerations, we think that our revised manuscript meets all the requirements of scientific novelty, technical robustness and general interest and impact for the research field, which makes it suitable for publication in *Nature Communications*.

Reviewers' Comments:

Reviewer #1:

Remarks to the Author:

The reply and modifications made in the text look quite convincing. I don't have further questions. I also read the replies to other referees and found them satisfactory. Therefore I recommend the paper for publications in Nature Communications.

Reviewer #2:

Remarks to the Author:

The authors have satisfactorily answered most of my questions. Below, I list remaining open questions towards the points (2) and (5) from my first report:

Ad 2: Indeed, Ref. [11] did not report a field-dependence of the NMR Knight shift, but recent a NMR work [PRX 9, 021044 (2019)] revealed a pronounced change of Knight shift in the normal state under uniaxial strain. My question is: can the local field (B_{loc}) variation indicate a change in susceptibility at the surface of SRO214 originating from local strain? The differences to the unstrained sample are most pronounced at low temperatures [Fig. 6 in PRX 9, 021044 (2019)], similar to the temperature range ($T < 25$ K) of the results in Fig. 3a of the present work. The authors should discuss such a possibility in the text (e.g. around line 132).

Ad 5: In addition to the scenarios discussed by the authors, there was a recent theoretical work [PRB 104, 024511 (2021)] suggesting that the signatures of time reversal symmetry breaking seen by μ -SR may originate from local strain inhomogeneities near edge dislocations. While in the superconducting state this may complicate the pairing towards a multi-component order parameter, such strain inhomogeneities are naturally relevant at the surface and should be mentioned in the present work. In particular, similar effects like those seen in the present μ -SR work on the sample's surface may occur inside the bulk nearby edge dislocations, albeit to a weaker extent than on the surface.

Once these remaining points are included, I recommend this manuscript for publication in Nature Communications.

Reviewer #3:

Remarks to the Author:

The authors have made substantial improvements to the manuscript and the underlying analysis. The paper is suitable for publication.

Referee 2

The authors have satisfactorily answered most of my questions. Below, I list remaining open questions towards the points (2) and (5) from my first report.

We thank Referee 2 for taking the time to review again our revised manuscript. We are glad that they have found our answers to their previous questions satisfactory.

Here, we address Referee 2's remaining open questions with a point-by-point response and explain further changes/additions that we have made to the manuscript to address fully any concerns.

- 1. Ad 2: Indeed, Ref. [11] did not report a field-dependence of the NMR Knight shift, but recent a NMR work [PRX 9, 021044 (2019)] revealed a pronounced change of Knight shift in the normal state under uniaxial strain. My question is: can the local field (B_{loc}) variation indicate a change in susceptibility at the surface of SRO₂₁₄ originating from local strain? The differences to the unstrained sample are most pronounced at low temperatures [Fig. 6 in PRX 9, 021044 (2019)], similar to the temperature range ($T < 25$ K) of the results in Fig. 3a of the present work. The authors should discuss such a possibility in the text (e.g. around line 132).*

We thank the referee for bringing the recent result of Y. Luo *et al.*, Phys. Rev. X 9, 021044 (2019) to our attention, in which O¹⁷ NMR measurements SRO₂₁₄ single crystals under strain are reported. We now cite this paper in the revised manuscript (Ref. 21) and discuss the paper below.

According to Y. Luo *et al.*, the Knight shift includes several contributions which cannot be fully separated, and it is specifically measured for the oxygen sites, whilst the muon stopping sites in our manuscript do not only coincide with oxygen sites, meaning that the interaction of the LE muons with the SRO₂₁₄ is fundamentally different. Also, in our study the Fermi level of the surface layers of SRO₂₁₄ is not at the van Hove singularity (VHS) and the layers below are bulk-like, as evidenced by previous angle-resolved photoemission spectroscopy measurements (see, for example, C. N. Veenstra *et al.*, Phys. Rev. Lett. 110, 097004 (2013)); on the other hand, in the NMR experiment of Y. Luo *et al.*, the authors reach the VHS through the application of uniaxial strain. These factors and the other factors discussed below make it challenging to draw a *quantitative* comparison between the local field enhancement (B_{loc}), which we observe at the SRO₂₁₄ surface by LE- μ SR, and the Knight shift results reported by Y. Luo *et al.* Nevertheless, we agree that there might be a correlation between the two experimental results. This possibility is also considered by Y. Luo *et al.* who suggest that dipolar fields due to orbital currents are the main contribution to the NMR Knight shift.

We first note that Y. Luo *et al.* report a paramagnetic shift with an anomaly due to spin fluctuations when the applied strain is close to the critical value ϵ_v , defined as the strain where the Fermi level crosses the VHS. The main experimental signatures discussed by Y. Luo *et al.* are therefore different from those that we associate to the orbital loop current phase on the SRO₂₁₄ surface, since this breaks time reversal symmetry, and it is static.

A further key difference which makes a comparison between the LE- μ SR shift in B_{loc} at the surface of SRO₂₁₄ and the NMR Knight shift difficult relates to the different magnitudes of the two signals as well as to different experimental conditions during measurement. For the NMR Knight shift results, Y. Luo *et al.* observe a shift of ~ 0.07 MHz, corresponding to a field of ~ 10 mT with an applied field of 8 Tesla. The surface enhancement in B_{loc} that we detect by LE- μ SR in an applied field of 0.15 Tesla (Fig. 3 of our manuscript) is less than 0.1% of the applied field meaning that the enhancement is of ~ 0.02 mT in applied field of 150 mT.

The B_{loc} shift at the SRO₂₁₄ surface, where the crystal structure differs from the bulk but not in an equivalent way to the crystal structure modification of SRO₂₁₄ under uniaxial strain, and the NMR shift therefore differ by several orders of magnitudes. In addition, given that we cannot apply magnetic fields much higher than those we used (0.15 T) and we cannot apply strain to the SRO₂₁₄ crystals in the LE- μ SR setup, it is difficult to determine if the shift in B_{loc} would increase using similar settings to the NMR measurements. For these reasons, we cannot conclude that B_{loc} due to orbital loop currents is enhanced in higher applied magnetic fields and so become a significant contribution to the NMR shift (which would be an essential result to draw a direct correlation between our results with Y. Luo *et al.* and to quantify the contribution of orbital loop currents to the NMR Knight shift under strain).

Although a direct quantitative comparison between our study and the NMR study is difficult to draw due to the different magnitudes of the two effects and measurement conditions, we cannot exclude that, near the critical strain ε_v , it is possible to induce in SRO₂₁₄ a similar phase to that which we probe at the SRO₂₁₄ surface. It is in fact possible that, through the application of ε_v , one could trigger fluctuations of the orbital loop current phase inside SRO₂₁₄. This picture is consistent with Y. Luo *et al*, where the Knight shift occurs at all the oxygen sites and not only at those sites affected by the VHS. Such behavior is explained by the authors in terms of a Stoner enhancement in the susceptibility signal, but their experimental observations are consistent with our above-proposed scenario, since the orbital currents flowing within a given RuO₄ plaquette also involve all the oxygen atoms within the plaquette.

Based on these considerations, we argue that the anomaly which Y. Luo *et al* report and relate to spin fluctuations could be related to an instability of the orbital current phase triggered when the strain approaches the critical value ε_v . In other terms, by applying a significant strain close to ε_v , it is possible that the SRO₂₁₄ system is driven into the same instability that we detect by muons on the SRO₂₁₄ surface, which is still structurally different from the structure of SRO₂₁₄ under strain, and that this state can manifest through an anomalous enhancement in the NMR shift.

We now discuss these points **on page 7 line 150** of the revised manuscript, where we have added the following text:

“We note that a paramagnetic Knight shift in the ¹⁷O nuclear magnetic resonance (NMR) signal has been recently measured for SRO₂₁₄ in its normal state under uniaxial strain²¹. The NMR Knight shift is of ~ 100 Gauss in an applied field of 8 x 10⁴ Gauss, and it exhibits an anomalous enhancement related to spin fluctuations at the critical strain ε_v , defined as the strain value where the Fermi level reaches the Van Hove singularity (VHS). We note that in our experiment, the Fermi level of the SRO₂₁₄ surface layers is not at the VHS and the layers underneath are just bulk-like as demonstrated by previous angle-resolved photoemission spectroscopy (ARPES) measurements^{22,23}, whilst in ref. ²¹ the authors reach the VHS through the application of ε_v . Despite these dissimilarities between the two experiments, one can argue that the surface of SRO₂₁₄ has a different local strain compared to the bulk, meaning that there may exist a correlation between our ΔB_{loc} enhancement at the SRO₂₁₄ surface and the Knight shift. Drawing a quantitative comparison between our ΔB_{loc} enhancement and Knight shift, however, is difficult for several reasons. First, as discussed in ref. ²¹, the Knight shift includes several contributions which cannot be fully separated and it is specifically measured for the oxygen sites, whilst the muon stopping sites do not simply coincide with the oxygen sites, meaning that the interaction of the muons with SRO₂₁₄ is different. Second, our ΔB_{loc} shift (~ 0.2 Gauss in a field of 1500 Gauss) is smaller than the Knight shift reported in ref. ²¹ by several orders of magnitudes, and it is measured in different experimental conditions from those of the NMR experiment, which cannot be reproduced in the LE- μ SR setup where neither larger magnetic field than those used, nor strain can be applied. These factors make it difficult to determine if and to which extent the shift in ΔB_{loc} would increase if the LE- μ SR measurements could be done using similar settings to the NMR measurements. Last, even if we cannot exclude that a correlation between the NMR Knight shift and ΔB_{loc} exists, our LE- μ SR measurements suggest that the ΔB_{loc} is characterized by different experimental signatures from those reported in ref. ²¹ for paramagnetic Knight shift because ΔB_{loc} originates from an ordered phase that breaks time reversal symmetry and that is also static in nature, as evidenced by our ZF measurements reported below.”

2. Ad 5: *In addition to the scenarios discussed by the authors, there was a recent theoretical work [PRB 104, 024511 (2021)] suggesting that the signatures of time reversal symmetry breaking seen by mu-SR may originate from local strain inhomogeneities near edge dislocations. While in the superconducting state this may complicate the pairing towards a multi-component order parameter, such strain inhomogeneities are naturally relevant at the surface and should be mentioned in the present work. In particular, similar effects like those seen in the present mu-SR work on the sample's surface may occur inside the bulk nearby edge dislocations, albeit to a weaker extent than on the surface.*

We agree that this other possible scenario should be considered. We now include this paper as **Ref. 47** in the revised manuscript and discuss this other scenario that may, in addition to the surface orbital loop currents, account for further discrepancies between the superconducting order parameter at the surface and bulk of SRO₂₁₄.

We now discuss this point **on page 15 line 362** of the revised manuscript, where we have added the following text:

“Dipolar fields generated near edge dislocations, which are particularly relevant near the SRO₂₁₄ surface due to local strain inhomogeneities, can also be a source of time reversal symmetry breaking⁴⁷ and therefore further contribute to the discrepancy in the symmetry of the superconducting order parameter determined based on bulk- and surface-sensitive spectroscopy techniques.”

3. Once these remaining points are included, I recommend this manuscript for publication in Nature Communications.

We have carefully addressed all the referee’s remaining remarks, which has contributed to further improving the quality of our manuscript. We believe that the referee will find our paper now suitable for publication in *Nature Communication*.

Reviewers' Comments:

Reviewer #2:

Remarks to the Author:

The authors have answered my remaining open questions in minute detail and adequately incorporated the related discussion in the manuscript. I support publication of the paper in Nature Communications. I am sure the community will be intrigued by these interesting findings.

Referee 2

The authors have answered my remaining open questions in minute detail and adequately incorporated the related discussion in the manuscript. I support publication of the paper in Nature Communications. I am sure the community will be intrigued by these interesting findings.

We thank Referee 2 for taking the time to review the second revised version of our manuscript. We are glad that they have found our answers to their previous questions satisfactory and that they recommend now the paper for publication in *Nature Communications*.

We think that the Referee 2's comments, together with those made by the other two Referees during the first round of review, have significantly improved the quality of our manuscript, for which we again express our sincere gratitude. We do hope that our results will stimulate a lot of discussions and interest in the scientific community, as both we and the Referees believe.